EMBO
Molecular Medicine

# *HUWE1* is a critical colonic tumour suppressor gene that prevents MYC signalling, DNA damage accumulation and tumour initiation

Kevin B Myant[1,2,*] , Patrizia Cammareri[1], Michael C Hodder[1], Jimi Wills[2], Alex Von Kriegsheim[2], Balázs Győrffy[3,4], Mamun Rashid[5], Simona Polo[6], Elena Maspero[6], Lynsey Vaughan[7], Basanta Gurung[8], Evan Barry[8], Angeliki Malliri[7], Fernando Camargo[8], David J Adams[5], Antonio Iavarone[9], Anna Lasorella[10] & Owen J Sansom[1,**]

## Abstract

Cancer genome sequencing projects have identified hundreds of genetic alterations, often at low frequencies, raising questions as to their functional relevance. One exemplar gene is *HUWE1*, which has been found to be mutated in numerous studies. However, due to the large size of this gene and a lack of functional analysis of identified mutations, their significance to carcinogenesis is unclear. To determine the importance of *HUWE1*, we chose to examine its function in colorectal cancer, where it is mutated in up to 15 per cent of tumours. Modelling of identified mutations showed that they inactivate the E3 ubiquitin ligase activity of HUWE1. Genetic deletion of *Huwe1* rapidly accelerated tumourigenic in mice carrying loss of the intestinal tumour suppressor gene *Apc*, with a dramatic increase in tumour initiation. Mechanistically, this phenotype was driven by increased MYC and rapid DNA damage accumulation leading to loss of the second copy of *Apc*. The increased levels of DNA damage sensitised *Huwe1*-deficient tumours to DNA-damaging agents and to deletion of the anti-apoptotic protein MCL1. Taken together, these data identify *HUWE1* as a *bona fide* tumour suppressor gene in the intestinal epithelium and suggest a potential vulnerability of *HUWE1*-mutated tumours to DNA-damaging agents and inhibitors of anti-apoptotic proteins.

**Keywords** colorectal cancer; DNA damage; HUWE1; MCL1; MYC
**Subject Categories** Cancer; Digestive System

## Introduction

The sequencing of human cancer genomes has led to a paradigm shift in our understanding of oncogenesis (Vogelstein *et al*, 2013). These studies have identified hundreds of genetic alterations that broadly segregate into two distinct groups, a small number of "mountains" (genes commonly mutated) and a large number of "hills" (gene mutated at low frequency). Whereas the causative role of frequently mutated genes is often clear, the role of those less commonly mutated can be difficult to separate from mutational "noise". Determining the relevance of low-frequency mutations is important for providing a comprehensive understanding of the processes driving tumourigenic. To date, most attempts at determining the relevance of such mutations have relied on computational approaches, mining large multi-cancer databases (Alexandrov *et al*, 2013; Lawrence *et al*, 2014). Whilst important, these approaches provide only correlative evidence and, as such, direct, functional testing remains the key determinant of the oncogenic potential of somatic mutations (Kadoch & Crabtree, 2013; Lewis *et al*, 2013).

*HUWE1* is an X-linked E3 ubiquitin ligase mutated at moderate frequencies (up to 15%) in a wide range of cancers including colorectal, uterine, gastric, cervical, melanoma and lung (Hodis *et al*, 2012; TCGA, 2012, 2014). HUWE1 catalyses the attachment of both lysine 48 (K48)- and lysine 63 (K63)-linked polyubiquitin chains, impacting on the function of a number of proteins involved in tumourigenic. The outcome of HUWE1-mediated K48 and K63 ubiquitination is quite different. For example, HUWE1 regulates the

1 Cancer Research UK Beatson Institute, Garscube Estate, Bearsden, Glasgow, UK
2 Cancer Research UK Edinburgh Centre, The Institute of Genetics and Molecular Medicine, Western General Hospital, Edinburgh, UK
3 MTA TTK Lendület Cancer Biomarker Research Group, Budapest, Hungary
4 2nd Department of Pediatrics, Semmelweis University, Budapest, Hungary
5 Wellcome Trust Sanger Institute, Hinxton, Cambridge, UK
6 IFOM, The FIRC Institute for Molecular Oncology, Milano, Italy
7 Cancer Research UK Manchester Institute, The University of Manchester, Withington, Manchester, UK
8 Boston Children's Hospital, Boston, MA, USA
9 Departments of Neurology and Pathology, Institute for Cancer Genetics, Irving Comprehensive Research Center, New York, NY, USA
10 Departments of Pediatrics and Pathology, Institute for Cancer Genetics, Irving Comprehensive Research Center, New York, NY, USA
*Corresponding author. Tel: +44 131 651 8635; E-mail: kevin.myant@igmm.ed.ac.uk
**Corresponding author. Tel: +44 141 330 3953; E-mail: o.sansom@beatson.gla.ac.uk

stability of MCL1 and TP53 via addition of K48-linked polyubiquitin chains (Chen *et al*, 2005; Zhong *et al*, 2005). Additionally, two recent studies have described a role for HUWE1 in DNA damage response and the regulation of genomic stability, again via K48-linked modulation of H2AX and PCNA stability (Atsumi *et al*, 2015; Choe *et al*, 2016). In contrast, HUWE1 regulates the function of DVL, a component of the WNT signalling pathway, via K63-linked ubiquitination (de Groot *et al*, 2014). This attachment prevents multimerisation of DVL, which is necessary for its role in binding AXIN2 during WNT signalling activation. Thus, HUWE1-mediated ubiquitination of DVL suppresses WNT activation. Perhaps most controversial is the role of HUWE1 in regulating MYC function. It has been reported that HUWE1 mediates K63 ubiquitination promoting the transcriptional activity of MYC, providing a pro-oncogenic function (Adhikary *et al*, 2005). However, HUWE1 has also been suggested to regulate the stability of both MYC during skin carcinogenesis and MYCN during brain development via K48 ubiquitination (Zhao *et al*, 2008; Inoue *et al*, 2013). Together, these studies produce contrasting predictions, one where HUWE1 may drive cancer via activation of MYC function and one where loss of HUWE1 would promote cancer by increasing levels of MYC, DNA damage and genomic instability. Thus far, evidence from cell line and xenograft studies exists to support both models, but genetic evidence from a skin cancer model showed *Huwe1* deletion accelerated tumourigenic suggesting a tumour suppressor role (Inoue *et al*, 2013). Mechanistically, the authors report *Huwe1* deletion leading to accumulation of the MYC/MIZ1 complex which, via direct promoter binding, suppresses expression of the anti-proliferative *P21* and *P15* genes (Inoue *et al*, 2013). Thus, in this model *Huwe1* suppresses tumourigenic primarily via anti-proliferative effects. However, the relevance of this model to human cancer is unclear as *HUWE1* mutations have not yet been observed in the currently limited, cutaneous squamous cell carcinoma sequencing studies (Lee *et al*, 2014).

Colorectal cancer (CRC), the second most common cause of cancer-related mortality, is a disease characterised by WNT signalling activation. Around 80% of tumours carry inactivating mutations in the *APC* gene, a key inhibitor of the WNT pathway. These mutations lead to a deregulation of WNT signalling that drives transformation of the intestinal epithelium. Particularly pertinent are our previous studies that show that *Myc* (a WNT target gene) and its downstream signalling targets are essential for the phenotype of deletion of *Apc in vivo* (Sansom *et al*, 2007; Myant *et al*, 2013; Faller *et al*, 2015). Indeed, haploinsufficiency for *Myc* can reduce the phenotypes of *Apc* loss and slow tumourigenic (Athineos & Sansom, 2010). This, in concert with the most frequent mutation of *HUWE1* in CRC, makes it an ideal model to characterise *HUWE1* function. Here, we robustly characterise the role of HUWE1 in CRC initiation. We find inactivating *HUWE1* mutations in human CRC and that deletion of *Huwe1* in CRC mouse models leads to rapid tumourigenic and hugely increased tumour initiation. MYC protein levels are increased and drive increased tumour proliferation but are not the primary cause of the increased tumour initiation. Rather, accumulation of DNA damage, characterised by accumulation of $\gamma$-H2AX leading to accelerated *Apc* loss, promotes tumour initiation. These tumours display increased sensitivity to DNA-damaging agents and are dependent on high levels of MCL1 for their survival

suggesting a potential therapeutic vulnerability of *HUWE1*-mutated tumours. Together, these data define *HUWE1* as a critical intestinal tumour suppressor gene that restrains cellular proliferation and DNA damage accumulation.

# Results

### HUWE1 is a colonic tumour suppressor

HUWE1 is a pleiotropic E3 ubiquitin ligase that modulates the function of several proteins involved in oncogenesis and DNA damage response including MYC, MYCN, MCL1 and H2AX (Adhikary *et al*, 2005; Zhong *et al*, 2005; Zhao *et al*, 2008; de Groot *et al*, 2014). Previous sequencing studies have identified somatic mutations throughout the *HUWE1* gene in up to 15% of colorectal tumours (Wood *et al*, 2007; Seshagiri *et al*, 2012; TCGA, 2012; Fig EV1A). *HUWE1* is a large gene (~15,000-bp cDNA), and thus, its frequent mutation could simply be a product of mutational "noise". *Huwe1* was also identified as a high-ranking positive hit in a CRC transposon mutagenesis screen (rank 66/752) indicating its mutation may be important to colorectal tumourigenic, but to date direct functional determination of this is lacking (March *et al*, 2011). Among many cancer types, CRC harbours the highest frequency of *HUWE1* mutations (Fig EV1B). However, the consequence of these mutations on the function of HUWE1 remains undetermined. To decipher the functional significance of *HUWE1* mutations, we interrogated the activity of two CRC-specific mutations targeting the HECT domain of *HUWE1* (R4082H and K4204del; Wood *et al*, 2007; Jones *et al*, 2008). Following expression and purification of the wild-type and mutant HECT domains of HUWE1, we found that both mutations led to marked inhibition of the ability of the HUWE1-HECT domain to bind to the E2-ubiquitin-conjugating enzyme UbcH7, thus indicating that the two mutants perturb the assembly of the HUWE1 ubiquitin ligase complex (Fig EV1C). As *HUWE1* is an X-linked gene that is transcriptionally silenced on the inactive X, mutation of a single allele would be sufficient to disrupt its activity (Carrel & Willard, 2005). Thus, we have identified potentially functionally inactivating mutations of *HUWE1* in primary colorectal tumours. We next addressed the consequence of *HUWE1* loss of function upon intestinal tumour development. We generated a cohort of mice carrying an inducible floxed allele of $Apc^{580S}$ (from here on referred to as $Apc^{fl}$) under the control of the *villin-Cre-ER*$^{T2}$ transgene (*vil-Cre-ER*$^{T2}$ $Apc^{fl/+}$— *Vil Apc*). Following recombination, these mice lose the wild-type *Apc* allele spontaneously and succumb to small intestinal and colonic tumourigenic. We crossed these mice to those carrying a conditional deletion allele of *Huwe1* ($Huwe1^{fl}$) to generate experimental cohorts of both female *vil-Cre-ER*$^{T2}$ $Apc^{fl/+}$ $Huwe1^{fl/+}$ (*Vil Apc Huwe1*$^{het}$) and female or male *vil-Cre-ER*$^{T2}$ $Apc^{fl/+}$ $Huwe1^{fl/fl}/Huwe1^{fl/y}$ mice (*Vil Apc Huwe1*$^{hom}$), respectively. Following Cre induction with tamoxifen, we aged these mice until signs of intestinal tumourigenic became apparent (pale feet, hunching and weight loss). Whereas control mice survived to a median of ~250 days, both *Huwe1* mutant cohorts succumbed to rapid tumourigenic (*Huwe1*$^{het}$ ~140 days, *Huwe1*$^{hom}$ ~90 days; Fig 1A). Strikingly, macroscopic and microscopic analysis of the guts from sacrificed mice indicated a dramatic increase in tumour number upon *Huwe1* deletion (Figs 1B–D and EV1D). Unlike *Vil*

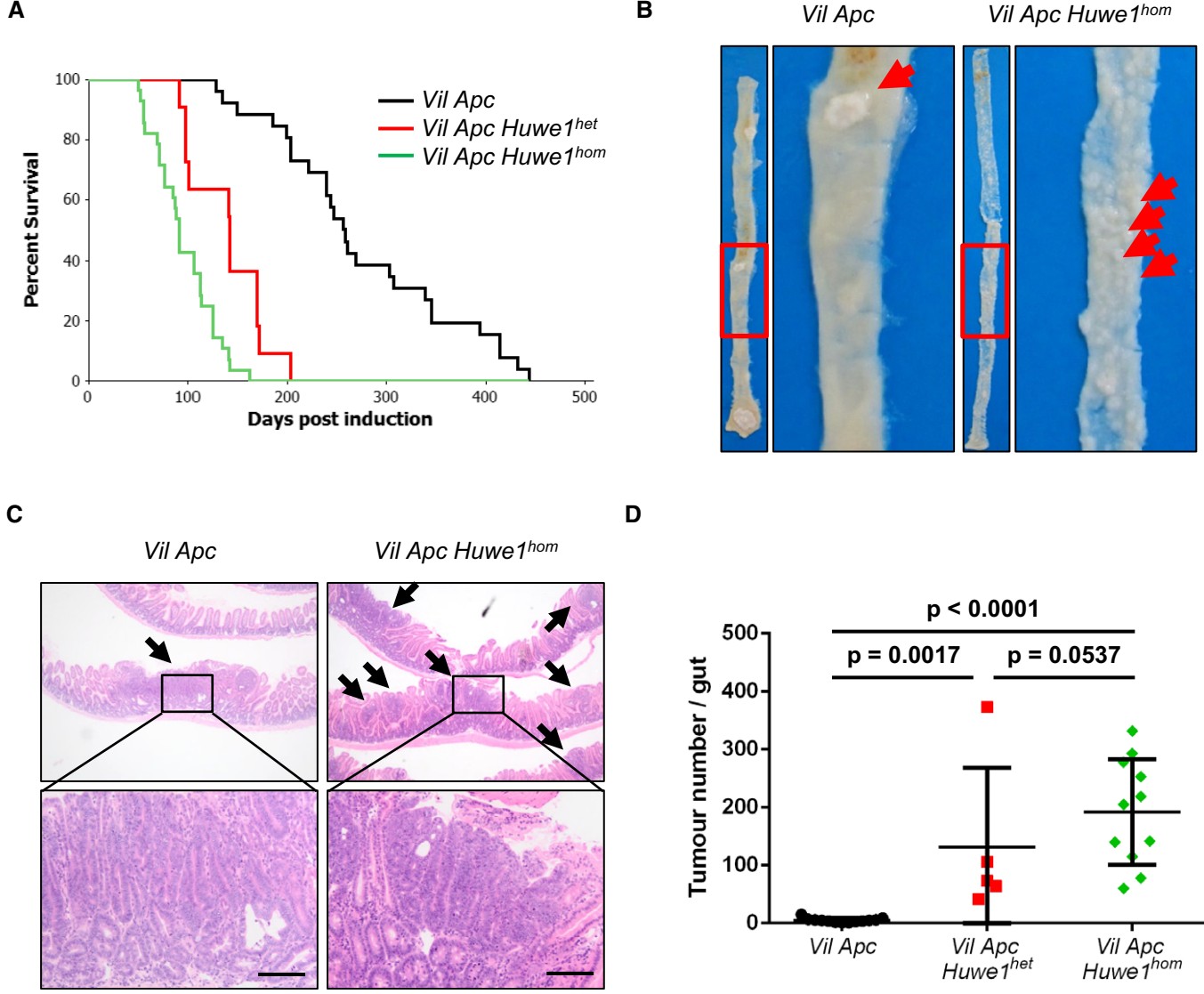

**Figure 1. Huwe1 is an intestinal tumour suppressor gene.**

A   Kaplan–Meier survival plot of cohorts of induced *Vil Apc*, *Vil Apc Huwe1^het^* and *Vil Apc Huwe1^hom^* mice. Deletion of *Huwe1* led to a significant reduction in survival of these animals (*Vil Apc* versus *Vil Apc Huwe1^het^/Vil Apc Huwe1^hom^*, log rank, P < 0.001, n ≥ 10).

B   Wholemount isolation of small intestines from *Vil Apc* and *Vil Apc Huwe1^hom^* mice culled at clinical endpoint. Note the huge numbers of tiny macroscopic adenomas visible in the *Vil Apc Huwe1^hom^* intestine (red arrows).

C   H&E staining of intestinal cross sections from *Vil Apc* and *Vil Apc Huwe1^hom^* mice demonstrating the increased adenoma burden following *Huwe1* deletion. Black arrows indicate individual adenomas. Scale bars = 200 μm.

D   Quantification of total tumour numbers per gut in sacrificed *Vil Apc*, *Vil Apc Huwe1^het^* and *Vil Apc Huwe1^hom^* mice. Deletion of *Huwe1* led to a significant increase in the number of tumours per gut (Mann–Whitney, n ≥ 5). Mean and standard deviation are plotted.

*Apc* mice that developed around five tumours, those from both the *Vil Apc Huwe1^het^* and *Vil Apc Huwe1^hom^* cohorts developed around 200 (Fig 1D). This significant increase in tumour number was also observed in the colons of homozygous deleted animals (Fig EV1D). Immunohistochemical analysis of these tumours demonstrated nuclear accumulation of β-catenin in *Huwe1^hom^* tumours at a similar level as control tumours indicating that tumour initiation was due to activation of WNT signalling (Fig EV2A–C). These data define *Huwe1* as an intestinal and

colonic tumour suppressor in the context of *Apc* heterozygosity whose loss of function leads to increased tumour initiation.

### Huwe1 deletion leads to perturbed intestinal homoeostasis

Given this profound impact on tumour formation, we next analysed whether *Huwe1* deletion impacted on intestinal homoeostasis. Histological examination of *Huwe1*-deficient intestines 14 days post-induction showed modest changes in crypt/villus architecture with

slightly shortened villi, but both cellular proliferation and apoptosis were unchanged (Fig 2A and B, and Appendix Figs S3A and S6A). We next analysed whether *Huwe1* loss altered intestinal cellular differentiation by examining markers of different intestinal cell populations. Interestingly, whilst we observed no gross changes in the number of goblet cells by periodic acid–Schiff (PAS) staining (Fig 2C), we observed lysozyme expression away from the intestinal crypt base (Fig 2D and Appendix Fig S3B and C). This could indicate either mislocalisation of Paneth cells or perturbed secretory intestinal differentiation. To investigate this more closely, we carried out double staining with PAS and alcian blue which marks Paneth cell secretory vesicles and goblet cells. This demonstrated that Paneth cell secretory vesicles are maintained at the crypt base in *Huwe1*-deficient intestines, suggesting that the lysozyme-positive cells may be cells from a different lineage expressing lysozyme (Fig 2E). Additionally, immunohistochemical analysis indicated that expression of the Paneth cell marker MMP7 was retained at the crypt base following HUWE1 deletion (Fig 2F). Furthermore, under closer scrutiny the lysozyme-positive cells further up the crypt–villus axis appear to have the morphology of goblet cells (Fig EV3D) suggesting alterations in secretory cell marker expression. Both Paneth cell differentiation (Andreu *et al*, 2008) and EPHB/EPHRINB gradient-mediated localisation (Batle *et al*, 2002; Sansom *et al*, 2004) are controlled by WNT signalling. A recent study has suggested a role for HUWE1 as a negative regulator of WNT signalling (Dominguez-Brauer *et al*, 2016), so we analysed the expression of a number of WNT target genes 14 days after *Huwe1* deletion. Interestingly, we observed a modest increase in expression of a number of WNT target genes, including *Ephb3*, which is a critical mediator of Paneth cell localisation (Fig EV3E; Batle *et al*, 2002). However, not all of the analysed WNT target genes showed increased expression, and once *Apc* was deleted, we no longer saw a HUWE1-dependent modification of WNT target genes, with the robust induction of targets caused by *Apc* loss masking any impact of HUWE1 (Fig EV3F). Thus, it is possible these changes reflect secondary consequences of *Huwe1* deletion effects on crypt homoeostasis. We next performed microarray analysis comparing wild-type to HUWE1-deficient intestines to assess whether WNT signalling was globally deregulated following *Huwe1* deletion. This was conducted at 4 days post-*Huwe1* deletion to reduce potential secondary consequences of *Huwe1* loss. We identified 586 and 415 genes whose expression was significantly up- or down-regulated, respectively (Appendix Table S1). The upregulated data set included a number of WNT-responsive intestinal stem cell marker genes including *Ascl2, Lect2, Slc14a1* and *Nrn1*. Gene set enrichment analysis identified a highly significant overlap of transcriptional markers of intestinal stem cells (chi-squared test with Yates's correction, $P < 0.0001$; Munoz *et al*, 2012) between the gene lists, demonstrating elevated expression of some members of the intestinal stem cell signature following *Huwe1* deletion. However, well-known WNT target genes such as *Axin2* and *Myc* were not upregulated and gross changes in nuclear β-catenin localisation were not observed (Fig EV3G), suggesting that HUWE1 loss only modifies the expression of subset of WNT target genes associated with stem cell markers. Together, these data identify a role for HUWE1 loss in moderate amplification of intestinal WNT signalling, expression of some stem cell-related genes and perturbed homoeostasis. However, given the redundancy of both stem cell markers (e.g. Lgr5) and Paneth cells

for intestinal homoeostasis and tumourigenic (de Lau *et al*, 2011; Durand *et al*, 2012), we felt these could not explain the marked increase in tumourigenic we observed.

## MYC drives increased tumour proliferation following *Huwe1* deletion

We next wanted to address the mechanism via which HUWE1 suppresses tumourigenic. Of particular relevance is the proposed role of HUWE1 in regulating MYC, a key downstream mediator of HUWE1 function (Inoue *et al*, 2013). As MYC is a critical modulator of intestinal tumourigenic, we hypothesised that the modulation of MYC stability may be a key tumour-suppressive function of HUWE1 within the gut (Sansom *et al*, 2007; Athineos & Sansom, 2010). We addressed this by determining MYC expression in epithelial extractions of control and *Huwe1*-deficient intestines. We found a significant increase in MYC protein, but not transcript, following HUWE1 loss (Fig 3A and B). This increase in MYC protein was also observed in *Huwe1*-deficient *Vil Apc Huwe1* tumours compared to controls and was independent of transcriptional effects (Fig 3C–E). These data indicate that HUWE1 controls MYC protein abundance in both normal intestine and intestinal tumours.

We next sought to determine whether MYC function is important for driving the dramatic phenotypes we observe upon *Huwe1* deletion. This was determined using both loss- and gain-of-function alleles of *Myc*. As *Myc* deletion rescues the phenotypes of *Apc* loss in the small intestine, complete ablation of *Myc* in the context of additional *Huwe1* deletion would likely yield the same phenotype. Moreover, as *Myc* haploinsufficiency slows *Apc*-dependent tumourigenic, one would expect it to also slow the *Vil Apc Huwe1* phenotype. Therefore, we wanted to find a system where tumourigenic is insensitive to a 50% reduction in *Myc* to permit assessment of the role of *Huwe1* in this context. In CRC, mutation and epigenetic inactivation of the PI3K–PTEN pathway occur in up to 40% of cases, and we have shown that *Pten* deletion rapidly accelerates tumourigenic and drives tumour progression (Marsh *et al*, 2008). *Pten* heterozygosity, however, only causes a relatively mild increase in tumourigenic. We therefore thought this model would be an ideal candidate to examine (i) whether *Huwe1* loss can accelerate tumourigenic in multiple spontaneous models and (ii) whether *Myc* is functionally important for this. We first confirmed that deletion of *Huwe1* accelerates tumourigenic in a spontaneous model with additional *Pten* deletion by ageing cohorts of *vil-Cre-ER^{T2} Apc^{fl/+} Pten^{fl/+}* (*Vil Apc Pten*) and *vil-Cre-ER^{T2} Apc^{fl/+} Pten^{fl/+} Huwe1^{fl/fl}* (*Vil Apc Pten Huwe1*) mice. Consistent with our previous model, *Huwe1* deletion led to a dramatic shortening of lifespan and a marked increase in tumour number and increased tumour cell proliferation (Figs 4A and B, and EV4A and B). We next determined whether this model is sensitive to reduced *Myc* expression by comparing survival of the *Vil Apc Pten* mice to those with additional deletion of a single *Myc* allele (*vil-Cre-ER^{T2} Apc^{fl/+} Pten^{fl/+} Myc^{fl/+}*—*Vil Apc Pten Myc*). Interestingly, we found neither survival of mice with *Pten* deletion nor tumour cell proliferation was sensitive to loss of a single *Myc* allele (Figs 4A and B, and EV4A and B). Therefore, this model is ideal for defining the functional relevance of elevated MYC levels in *Huwe1*-deficient tumours. We generated *vil-Cre-ER^{T2} Apc^{fl/+} Pten^{fl/+} Huwe1^{fl/fl} Myc^{fl/+}* (*Vil Apc Pten Huwe1 Myc*) mice and compared their survival to the *Vil Apc Pten Huwe1* cohort. Strikingly, in the

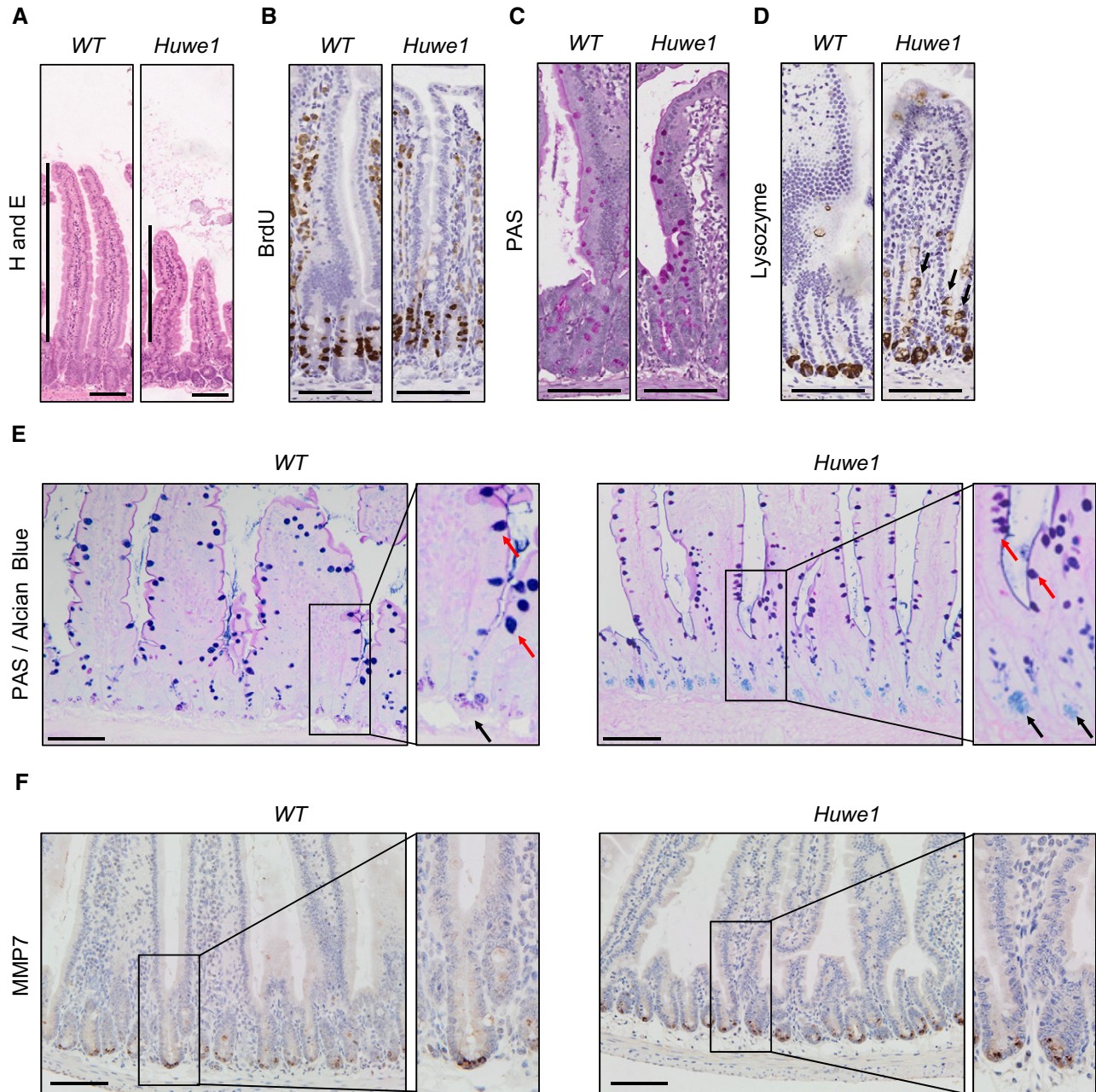

**Figure 2. Huwe1 deletion leads to perturbed intestinal homoeostasis.**

A  H&E staining of control and *Huwe1*-deleted intestinal epithelium. Shortened villi in *Huwe1*-deficient tissue are indicated.

B  BrdU IHC of control and *Huwe1*-deleted intestinal epithelium.

C  PAS staining identifying goblet cells. No gross changes were observed.

D  Lysozyme staining (Paneth cell marker) of control and *Huwe1*-deleted small intestine. Note the occurrence of lysozyme-positive cells away from the crypt base (black arrows).

E  Dual periodic acid–Schiff/alcian blue staining to identify Paneth cell secretory vesicles (light blue/pink, marked with black arrows) and goblet cells (dark blue/purple, marked with red arrows). Note Paneth cell secretory vesicles are restricted to crypt base in both control and *Huwe1*-deficient small intestines (inset, black arrows).

F  MMP7 staining of control and *Huwe1*-deleted small intestine. Note MMP7 staining is restricted to crypt base in *Huwe1*-deficient intestines.

Data information: Scale bars = 100 μm.

context of *Huwe1* deletion, reduced expression of *Myc* led to a significant increase in survival, thus indicating that the tumour suppressor function of HUWE1 is, at least in part, mediated through the control

of MYC stability (Fig 4A). However, despite the enhanced survival, *Vil Apc Pten Huwe1 Myc* mice showed no reduction in tumour number (Fig 4B). They did, however, show decreased tumour

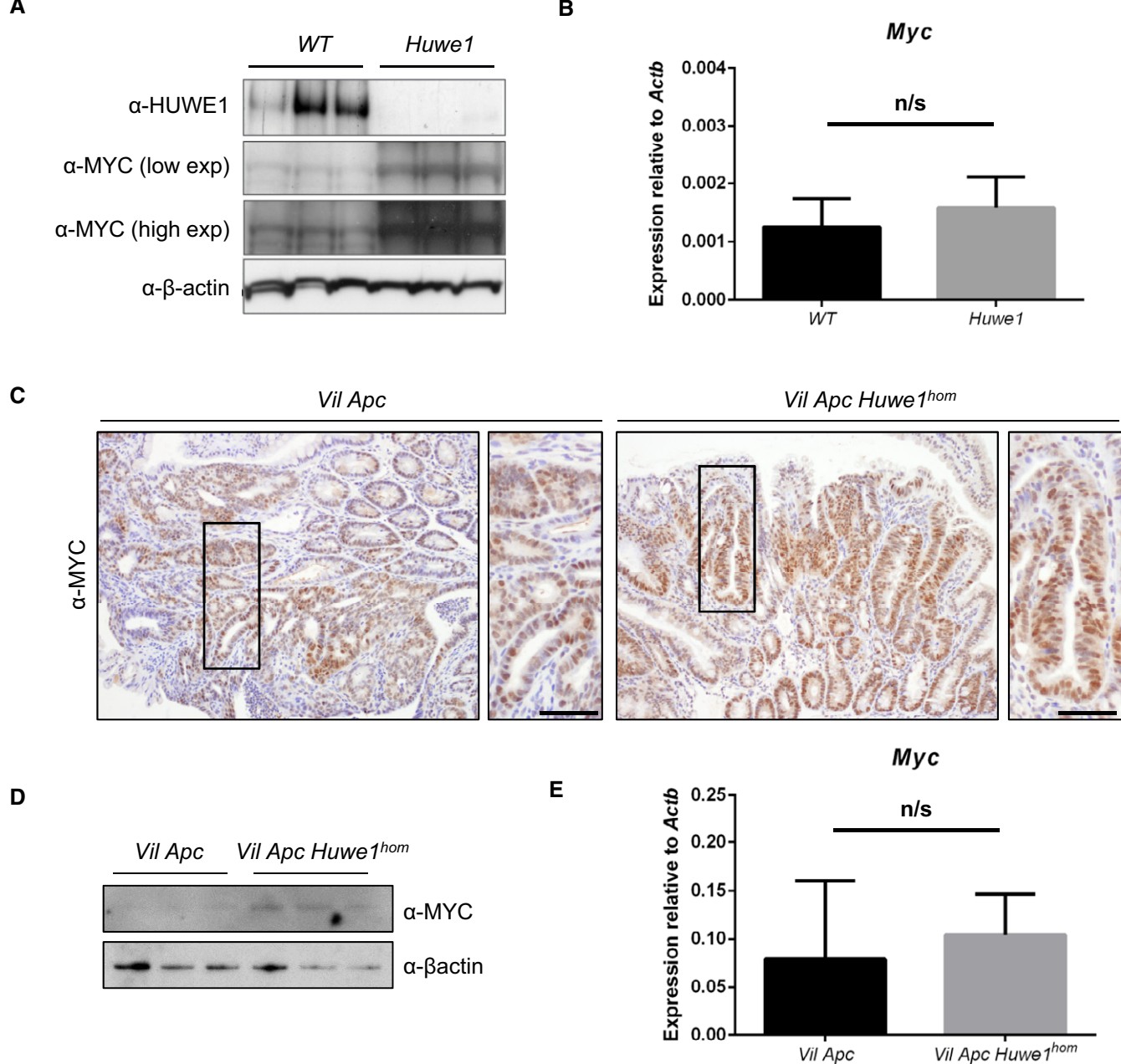

**Figure 3.  Increased MYC protein expression following *Huwe1* deletion.**

A   HUWE1 and MYC Western blot analysis of epithelial cell extractions from control and *Huwe1*-deficient intestines. Levels of MYC protein are significantly increased in normal intestines lacking HUWE1 (Mann–Whitney, $P = 0.04$, $n = 3$).

B   qRT–PCR analysis of *Myc* expression in control and *Huwe1*-deficient intestines. Mann-Whitney, $n = 4$ vs 6. Data plotted are mean and SD.

C   MYC IHC in tumours from *Vil Apc* and *Vil Apc Huwe1*^*hom*^ mice. Scale bars = 50 µm.

D   MYC Western blot in protein extracts from *Vil Apc* and *Vil Apc Huwe1*^*hom*^ tumours. Levels of MYC protein are significantly increased in tumours lacking HUWE1 (Mann–Whitney, $P = 0.04$, $n = 3$).

E   qRT–PCR analysis of *Myc* expression in tumours from *Vil Apc* and *Vil Apc Huwe1*^*hom*^ mice. Mann-Whitney, $n = 3$ vs 3. Data plotted are mean and SD.

Source data are available online for this figure.

proliferation rates compared to tumours from *Vil Apc Pten Huwe1* mice (Fig EV4A and B). These data suggest that MYC accumulation in this model promotes tumour growth, but the residual MYC levels in the *Myc*^*fl/+*^ background are sufficient for efficient tumour initiation. We further characterised the role of increased MYC protein during tumourigenic by overexpressing two copies of a proteolytically stable MYC mutant (*Rosa-lsl-Myc*^*T58A*^) in this model (*Apc Pten Myc*^*T58A*^). Consistent with our loss-of-function experiments,

overexpression of MYC[T58A] significantly decreased survival but had no effect on overall tumour number (Fig 4C and D). In human CRC, we found that *HUWE1* mutations were mutually exclusive with amplifications, gains and transcriptional upregulation of *MYC* ($P = 0.042$; Fig 4E). This suggests that *HUWE1* mutation in human CRC might facilitate increased levels of MYC and bypass the requirement for genomic amplification. However, it is also clear from our work, and others, that MYC levels are sufficiently high for transformation in the presence of HUWE1 and reduction or overexpression of MYC does not have a profound impact on tumour initiation.

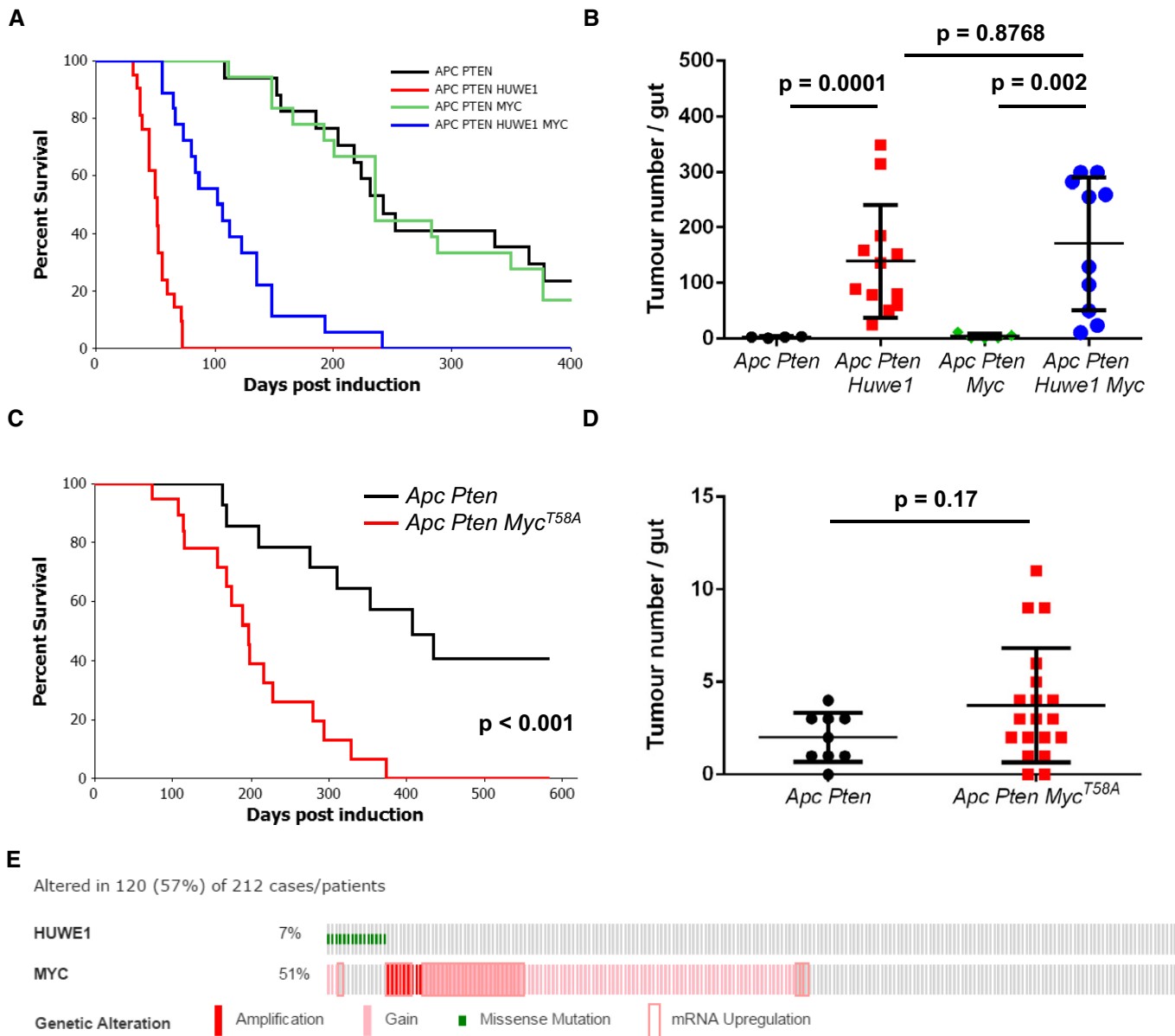

**Figure 4.  MYC upregulation following *Huwe1* deletion drives accelerated tumourigenic.**

A    Kaplan–Meier survival plot of cohorts of induced *Vil Apc Pten*, *Vil Apc Pten Huwe1*, *Vil Apc Pten Myc* and *Vil Apc Pten Huwe1 Myc* mice. Heterozygous deletion of *Myc* led to a specific and significant increase in survival of *Huwe1*-deleted animals (*Vil Apc Pten Huwe1* versus *Vil Apc Pten Huwe1 Myc*, log rank, $P < 0.001$, $n \geq 16$).

B    Quantification of total tumour numbers per gut in sacrificed *Vil Apc Pten*, *Vil Apc Pten Huwe1*, *Vil Apc Pten Myc* and *Vil Apc Pten Huwe1 Myc* mice. Heterozygous deletion of *Myc* did not reduce the number of tumours in *Huwe1*-deleted mice (*Vil Apc Pten Huwe1* versus *Vil Apc Pten Huwe1 Myc*, Mann–Whitney, $n \geq 10$). Data plotted are mean and SD.

C    Kaplan–Meier survival plot of cohorts of induced *AhCre-ER^T Apc Pten* and *AhCre-ER^T Apc Pten Myc^{T58A}* mice. Overexpression of proteolytically stabilised MYC led to a significant reduction in survival (log rank, $P < 0.001$, $n = 14$ versus 21).

D    Quantification of total tumour numbers per gut in sacrificed *Apc Pten* and *Apc Pten Myc^{T58A}* mice. Overexpression of proteolytically stabilised MYC did not increase the number of tumours in *Apc Pten* mice (Mann–Whitney, $n = 9$ versus 19). Data plotted are mean and SD.

E    cBioportal OncoPrint showing mutual exclusivity of *HUWE1* mutations and increased *MYC* copy number and/or RNA expression in CRC samples (log odds ratio: −1.144 [some tendency towards mutual exclusivity], $P = 0.042$).

Taken together, the above findings indicate that, besides MYC, other substrates of HUWE1 contribute to intestinal tumourigenesis in *Huwe1*-null mice.

### *Huwe1* deletion leads to increased levels of DNA damage

The massively increased tumour number we observed was reminiscent of mice displaying a mutator phenotype, such as those deficient for *Mlh1* or *Msh2* (Reitmair *et al*, 1996; Edelmann *et al*, 1999). We reasoned that increased tumour initiation in mice heterozygous for *Apc* loss would most likely occur via increased DNA damage and/or mutation rate. Two recent reports have described a role for HUWE1 in regulation of DNA damage response and genomic stability via degradation of H2AX and PCNA, respectively (Atsumi *et al*, 2015; Choe *et al*, 2016). To attempt to define the mechanism by which *Huwe1* deletion drives increased tumourigenic, we carried out mass spectrometry analysis to compare the proteome of normal and *Huwe1*-deficient intestinal tissue. Interestingly, this analysis identified a significant increase in H2AX levels following *Huwe1* deletion that we confirmed by Western blot (Appendix Table S2 and Fig 5A). Thus, loss of *Huwe1* function could lead to accumulation of DNA damage via accumulation of H2AX leading to inefficient resolution of γ-H2AX foci (Atsumi *et al*, 2015). To test this hypothesis, we analysed DNA damage levels following *Huwe1* deletion. Interestingly, we observed increased levels of γ-H2AX in normal intestine and tumour tissue deficient for *Huwe1* (Fig 5B–D and Appendix Fig S5A). We next used PCR analysis of the *Apc* locus to determine the mechanism by which *Huwe1*-deficient cells lose *Apc*. The majority of both control and *Huwe1*-deficient tumours demonstrated loss of heterozygosity at the *Apc* locus indicating that *Apc* loss is driving tumour initiation (Fig EV5B). A common property of cells with increased levels of DNA damage is an increased sensitivity to DNA-damaging agents (Farmer *et al*, 2005; Hay *et al*, 2005). To investigate this further, we treated both control and tumour-bearing mice with the DNA-cross-linking agent cisplatin. Consistent with the observed increase in DNA damage, *Huwe1*-deficient tumours displayed increased sensitivity to this treatment with significantly higher levels of apoptosis (Fig 6A and B). Together, these data demonstrate that loss of *Huwe1* leads to accumulation of DNA damage, loss of *Apc* and subsequent tumour initiation. We next analysed tumour data from TCGA to determine whether *HUWE1* loss of function can confer a similar DNA damage phenotype in human disease. We found that tumours carrying *HUWE1* mutations displayed a far higher somatic mutational burden than those with wild-type *HUWE1* (Fig 6C). Importantly, this increase was independent of MLH1 silencing, a key determinant of mutational burden indicating this is not merely a consequence of a general increased mutation rate (Fig 6D). Thus, increased DNA damage appears to be a property of *HUWE1* mutation in human CRC and may confer a therapeutic vulnerability on these tumours during tumourigenic. One prediction of increased sensitivity to DNA-damaging agents upon *HUWE1* loss would be that tumours that had low levels of HUWE1 might respond better to therapy. There are currently no data publically available on the treatment response of *HUWE1*-mutated patients, so we analysed this with respect to *HUWE1* expression levels. We found that patients whose tumours express low levels of *HUWE1* have significantly increased overall survival following chemotherapy treatment than those expressing higher

levels (Fig 6E). Interestingly, there were no significant survival differences between patients who received no adjuvant chemotherapy treatment based on *HUWE1* expression levels (Fig 6F). These data are consistent with an increased DNA damage burden in human tumours depleted of *HUWE1* and may suggest that patients with *HUWE1* mutations may respond better to chemotherapeutic treatment.

Given this increased DNA damage even in intestines that were wild-type for *Apc*, this could suggest that *Huwe1* loss increases tumour initiation by promoting loss of the wild-type *Apc* allele. To test whether *Huwe1* loss accelerates tumourigenic independently of the requirement for loss of wild-type *Apc*, we utilised a model initiated by homozygous deletion of *Apc*. Using the *Lgr5-cre-ER^T2* knock-in mouse, both copies of *Apc* alone or in combination with *Huwe1* can be deleted specifically in stem cells (Fig 7A). This model permits rapid tumourigenic originating from LGR5-positive intestinal stem cells. We generated cohorts of *Lgr5-cre-ER^{T2} Apc^{fl/fl}* (*Lgr5 Apc*) and *Lgr5-cre-ER^{T2} Apc^{fl/fl} Huwe1^{fl/fl}* (*Lgr5 Apc Huwe1*) mice and aged them until they showed signs of tumourigenic (Fig 7B). In this model, there was a less dramatic (median survival 50 versus 43 days) though significant decrease in survival of *Lgr5 Apc Huwe1* mice compared to *Lgr5 Apc* (*P* < 0.001; Fig 7B). This indicates that HUWE1 suppresses intestinal tumourigenic at multiple levels, in part by stopping tumour initiation but (and consistent with the data on MYC above) also at the level of tumour growth. To determine whether the lysozyme expression and stem cell phenotypes were maintained in tumour tissue, we analysed lysozyme and OLFM4 expression in these tumours. We observed both increased lysozyme-positive cell abundance and zone of OLFM4 expression in *Huwe1*-deficient tumours (Fig 7C and D). Thus, even in the absence of *Apc* loss of *Huwe1* perturbed differentiation.

One very surprising observation was that despite the dramatically increased DNA damage burden, we did not observe an increase in apoptosis in *Huwe1*-deficient crypts (Fig EV6A). Intestinal epithelial cells are extremely sensitive to DNA damage-induced apoptosis and equivalent levels of γ-H2AX would normally lead to high levels of apoptosis, suggesting that *Huwe1*-deficient cells are protected from cell death (Phesse *et al*, 2014). HUWE1 has previously been implicated in modulating the stability of the anti-apoptotic protein MCL1 and so we hypothesised that increased levels of MCL1 may protect *Huwe1*-deficient cells from death (Zhong *et al*, 2005). We observed increased levels of MCL1 in tumour tissue deficient for *Huwe1* (Fig 8A). To determine whether increased MCL1 levels protect *Huwe1*-deficient cells from apoptosis, we generated *VilCreERT2 Huwe1^{fl/fl} Mcl1^{fl/+}* (*Huwe1 Mcl1*) mice and induced simultaneous deletion of *Huwe1* and one copy of *Mcl1*. The intestines of these mice displayed significantly higher levels of apoptosis than singly deleted *Huwe1* mice demonstrating a role for MCL1 in protecting *Huwe1*-deficient cells from apoptosis in homoeostatic conditions (Fig EV6A). Importantly, deletion of one copy of *Mcl1* alone did not induce an apoptotic response suggesting this observation was specific to *Huwe1* loss-mediated DNA damage (Fig EV6A). We next addressed whether this impacted on tumour development by generating a cohort of tumour-inducible mice carrying the same *Mcl1^{fl/+}* allele. Similar to our observations in normal tissue, we found that heterozygous deletion of *Mcl1* reduced *Huwe1* loss-driven tumourigenic (Fig 8B). Mice heterozygous for *Mcl1* displayed a greater propensity to form indolent lesions than the mice wild-type

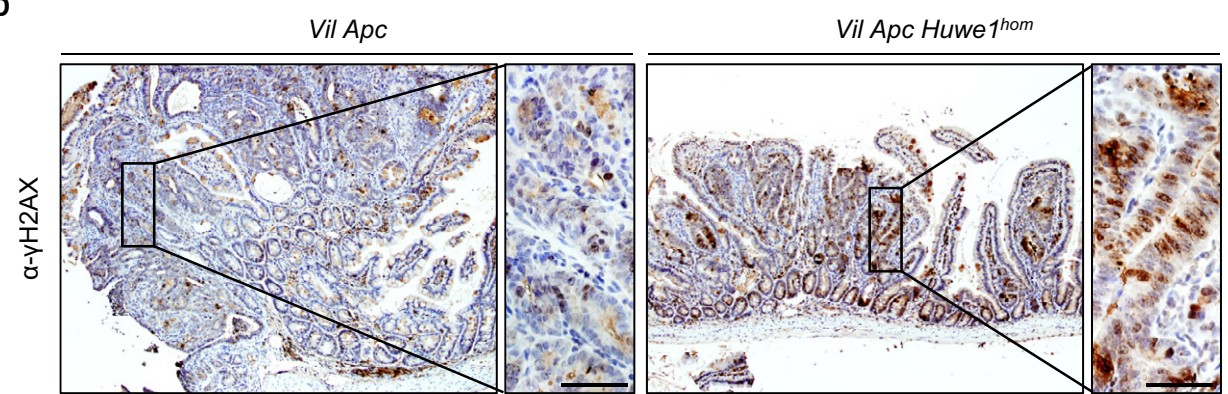

**Figure 5.  *Huwe1* deletion leads to an increase in DNA damage.**

A   H2AX Western blot in protein extracts from control and *Huwe1*-deficient intestinal epithelial cells. Band intensity relative to β-actin displayed under each lane (Mann–Whitney, $P = 0.04$, $n = 3$ versus 3).

B   γ-H2AX IHC showing increased positivity in *Huwe1*-deleted intestinal cells (black arrows). Scale bars = 50 μm.

C   γ-H2AX Western blot in protein extracts from control and *Huwe1*-deficient intestinal epithelial cells (Mann–Whitney, $P = 0.04$, $n = 3$ versus 3).

D   γ-H2AX IHC of *Vil Apc* and *Vil Apc Huwe1*[hom] tumours; note increased positivity in *Huwe1*-deficient tumours. Scale bars = 50 μm.

Source data are available online for this figure.

for *Mcl1*. The ratio of canonical adenomas to indolent lesions, identified as a spherical lesion usually contained entirely within a single villus, was significantly different between the genotypes (Fig 8C and Appendix Fig S6B and C). Together, these data indicate that *Huwe1* loss of function leads to increased levels of MCL1 protein and this protects cells from DNA damage-induced apoptosis facilitating oncogenic transformation.

## Discussion

The identification of hundreds of somatic mutations in cancer genomes has raised critical questions as to their functional relevance. Despite efforts to analyse such mutations computationally, our work demonstrates the importance of direct functional testing, in particular of large genes mutated at moderate levels. Our work shows, by robust genetic characterisation, that *Huwe1* is a tumour suppressor in the small intestine and colon. Together with our identification of *HUWE1* mutations present in human CRC that perturb

its ubiquitin ligase activity, this strongly suggests it is a *bona fide* colonic tumour suppressor gene.

This is particularly important as previous work on HUWE1's tumourigenic role had proven controversial. HUWE1 is an E3 ubiquitin ligase that controls the stability of MCL1, MYC and MYCN functions which would suggest a tumour-suppressive role. Indeed, work using chemically induced skin cancer mouse models has indicated this is the case (Inoue *et al*, 2013). However, via K63-mediated ubiquitination, HUWE1 is required for the transactivation function of MYC and tumour cell proliferation. It is also overexpressed in a number of different cancers including colorectal, supporting a pro-oncogenic function (Adhikary *et al*, 2005). Using intestinal and colonic specific gene deletion, we show that loss of *Huwe1* leads to significantly accelerated tumourigenic characterised by a massive increase in tumour incidence. We observe coincident increased MYC protein levels and, using conditional co-deletion of *Myc*, we demonstrate this increase in MYC is an important mediator of this phenotype. The mutual exclusivity of *HUWE1* mutation and *MYC* genomic amplification we observe in human CRC strongly

**A**

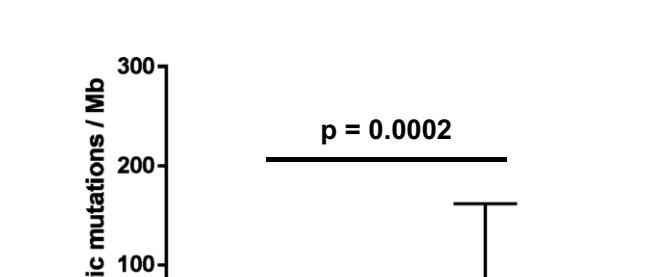

**B**

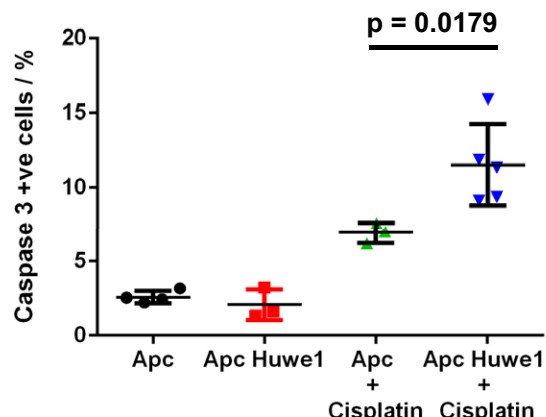

**C**

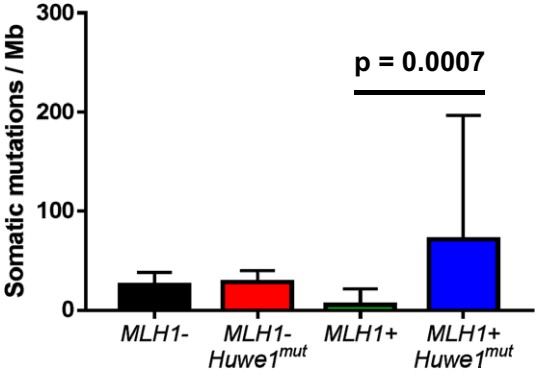

**D**

**E**

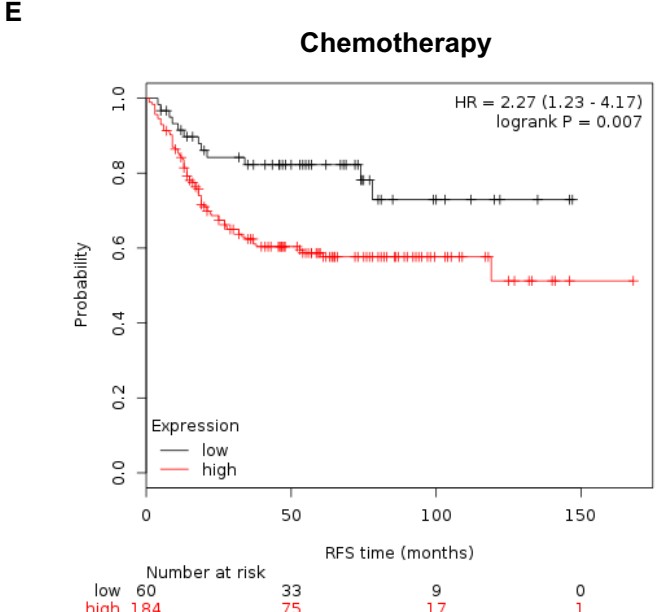

**F**

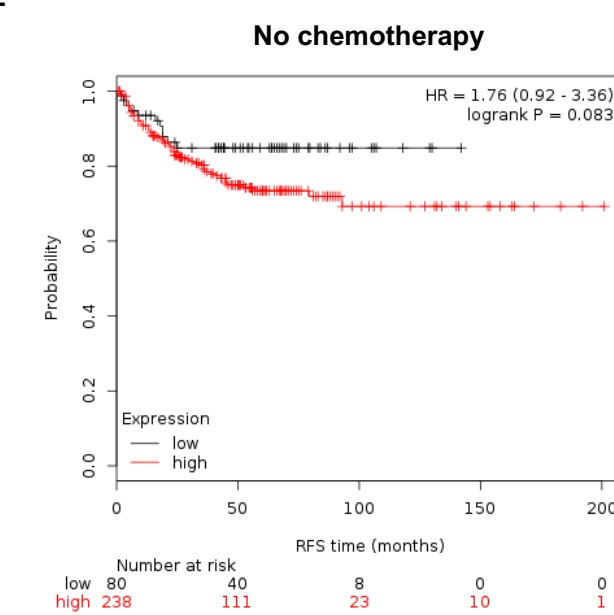

**Figure 6.**

**Figure 6. Huwe1-deficient tumours are sensitive to cisplatin treatment.**

A  Caspase-3 IHC of tumours from *Lgr5 Apc* and *Lgr5 Apc Huwe1* mice either untreated or treated with 7.5 mg/kg cisplatin 6 h post-treatment. Arrows identify caspase-3-positive cells. Scale bars = 50 μm.

B  Quantification of cisplatin treatment showing a significant increase in apoptosis in *Huwe1*-deficient tumour cells (Mann–Whitney, n = 3 versus 5). Data plotted are mean and SD.

C  Comparison of somatic mutation rate (mutations/Mb) between human tumours carrying *HUWE1* mutations or not. Note the significant increase in mutational burden in *HUWE1*-mutated tumours (Mann–Whitney, $P = 0.0002$, $n \geq 15$). Data plotted are mean and SD.

D  Comparison of somatic mutation rate (mutations/Mb) between human tumours carrying *HUWE1* mutations or not grouped according to MLH1 status. Note that the increased mutation rate is found primarily in tumours where MLH1 is not silenced indicating increased mutation rate (and *HUWE1* mutation itself) is not due to silencing of MLH1 (Mann–Whitney, $P = 0.0007$, $n \geq 11$). Data plotted are mean and SD.

E  Survival analysis of colorectal cancer patients treated with adjuvant chemotherapy divided by *HUWE1* expression levels. Note the lowest *HUWE1*-expressing quartile of patients respond significantly better to chemotherapy.

F  Survival analysis of colorectal cancer patients not treated with adjuvant chemotherapy divided by *HUWE1* expression levels. Note the lowest *HUWE1*-expressing quartile of patients do not survive significantly longer than those expressing higher levels of *HUWE1* if not treated with chemotherapy.

implicates them as critical mediators of colorectal tumourigenic. Thus, our findings are consistent with a tumour-suppressive role of HUWE1 in destabilising MYC post-translationally. The reason for these discrepancies is not immediately clear although it is worth noting the elevated levels of DNA damage we observed following *Huwe1* deletion *in vivo*. It is possible that some of the anti-proliferative effects the authors observe following *HUWE1* knockdown are in fact due to DNA damage-induced growth arrest that is more apparent in cell culture conditions. Interestingly, *Myc* deletion does not impact on the increased tumour initiation phenotype; rather, it dampens the elevated tumour cell proliferation observed in *Huwe1*-deficient tumours. This has the consequence of slowing the accelerated tumour growth following *Huwe1* loss but has no impact on the ability of tumours to form. Thus, HUWE1 clearly impacts on tumourigenic via multiple pathways.

Our observation that lysozyme expression becomes mislocalised and the stem/progenitor cell zone was expanded is consistent with a recent paper identifying HUWE1 in regulating the intestinal stem cell niche via direct modulation of EPHB3 stability (Dominguez-Brauer *et al*, 2016). However, we believe that the impact of Paneth cell marker mislocalisation due to increased EPHB3 stabilisation is insufficient to explain the tumourigenic phenotype observed in both studies. Firstly, although expression of lysozyme was mislocalised, Paneth cell secretory vesicles and MMP7 expression were not, suggesting this phenotype may be the consequence of perturbed lineage commitment rather than gross changes in Paneth cell localisation. In addition, previous studies have shown induction of Paneth cell mislocalisation via perturbation of EPHB/EPHRINB gradient leads to modest changes in tumourigenic with the most striking aspect being CRC progression. Of note, we did not observe any invasive carcinomas in our *Apc^{fl/+} Huwe1*-deficient mice. Rather, we observed an ~40-fold increase in tumour number together suggesting changes in EPHB3 stabilisation are not the primary cause of tumour initiation following *Huwe1* deletion. Notably, we also observed increased tumour formation initiated directly from LGR5-positive stem cells in *Huwe1*-deficient intestines. Using a stem cell-specific cre, we were able to show not only accelerated tumourigenic, but also maintenance of the lysozyme expression phenotype observed in normal tissue. As stem cells are proposed to be the cell of origin of colorectal cancer (Barker *et al*, 2009), the increased propensity of those lacking *Huwe1* to undergo transformation may in part explain its tumour-suppressive role. However, as yet there is no evidence that increased numbers of intestinal stem cells would lead to transformation. Indeed, one

might predict that increased stem cell number might increase stem cell competition, so the chance that a second mutation in APC is fixed is reduced (Vermeulen *et al*, 2013). Moreover, we did not observe tumour formation in intestines deficient for *Huwe1* alone, even a year post-induction suggesting that disruption of homoeostasis (e.g. deregulation of stem cell markers and Paneth cells) was not sufficient in driving carcinogenesis. The significance of previously reported roles of HUWE1 and WNT signalling is still to be determined. We only saw activation of a subset of ISC signature WNT targets and not general targets such as AXIN2, and there were no changes in nuclear β-catenin. More importantly, all tumours lost the second copy of *Apc*, and once this occurred, WNT target expression was similar between wild-type and Huwe1-deficient intestines. Together, this would suggest that neither the disruption of homoeostasis nor WNT signalling can explain the rapid tumourigenic observed. Thus, we think it is most likely that tumours arise due to different tumour-suppressive mechanisms exerted by HUWE1.

Particularly pertinent to this was our observation of high levels of DNA damage in *Huwe1*-deficient tissue and tumours. Our finding that, in addition to increased levels of DNA damage, *Huwe1*-deficient tumours display additional sensitivity to DNA-damaging agents is in agreement with previous reports (Farmer *et al*, 2005; Hay *et al*, 2005). A number of previous studies have identified HUWE1 as an important mediator of DNA repair via modulation of proteins such as MUTY, BRCA1, POLB and TP53 (Chen *et al*, 2005; Parsons *et al*, 2009; Dorn *et al*, 2014; Wang *et al*, 2014). Interestingly, alongside mediating DNA damage response, a recent paper has identified a mechanism via which *Huwe1* loss can *drive* DNA damage accumulation. Choe and colleagues reported HUWE1 promotes replication of damaged DNA via interaction with PCNA leading to efficient H2AX signalling (Choe *et al*, 2016). In the absence of *Huwe1*, DNA damage accumulates and, similar to our own findings, cells show increased sensitivity to DNA-damaging agents. This is in agreement with another study identifying H2AX as a direct target for HUWE1-mediated degradation (Atsumi *et al*, 2015) and our own findings that γ-H2AX accumulates in intestinal epithelial cells following *Huwe1* deletion. Together, these data suggest HUWE1 is a critical mediator of DNA damage response and repair and loss of *Huwe1* can drive DNA damage accumulation. They also indicate an additional important facet of *HUWE1*-mutated colorectal tumours—that they may be exquisitely sensitive to DNA-damaging agents and may therefore respond better to them. This is supported by our finding that patients whose tumours

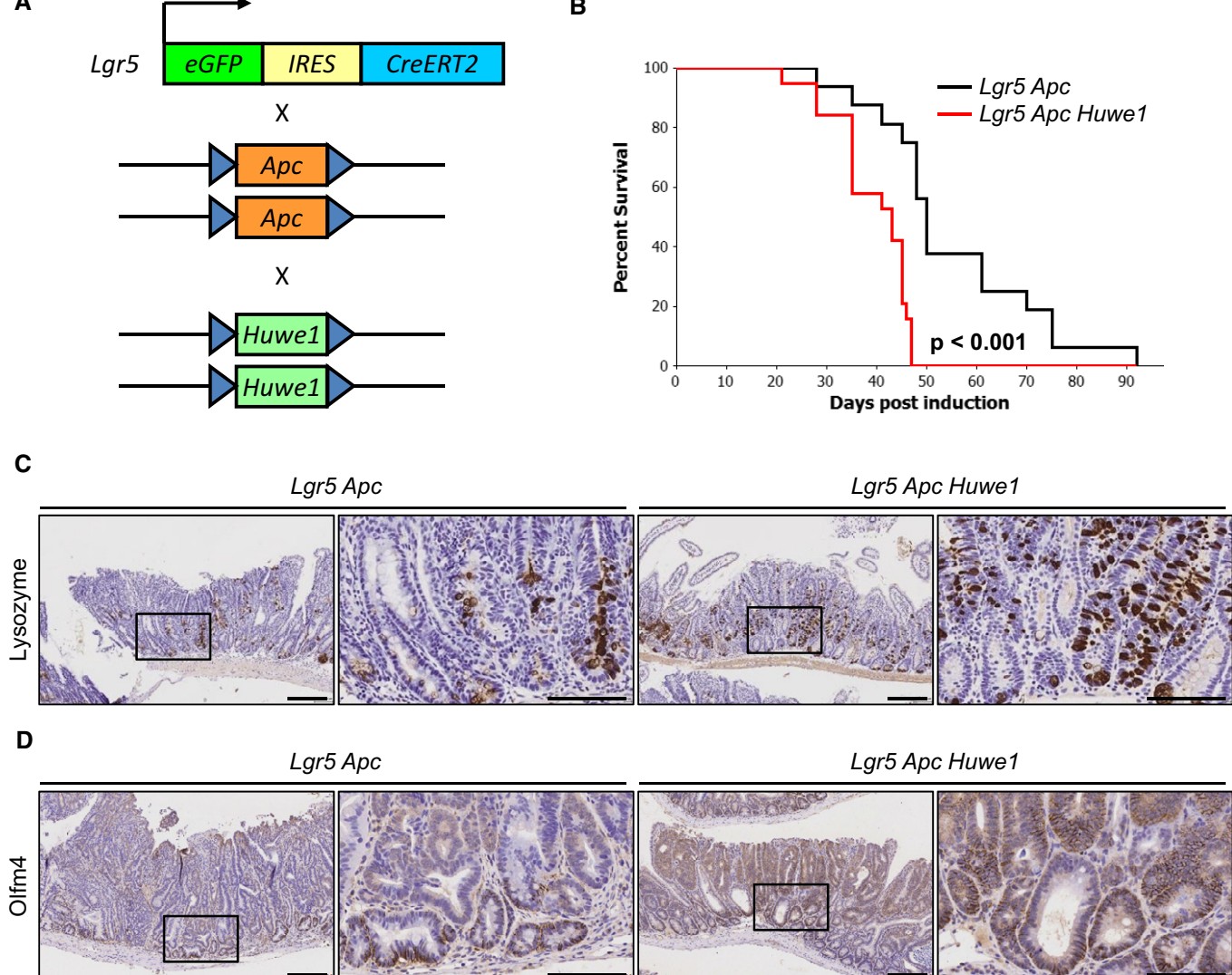

**Figure 7.  Huwe1 loss promotes intestinal stem cell transformation.**

A    Schematic outlining Lgr5 Apc Huwe1 stem cell transformation/tumour model.
B    Kaplan–Meier survival plot of cohorts of induced Lgr5 Apc and Lgr5 Apc Huwe1 mice. Deletion of Huwe1 led to a significant reduction in survival of these animals (log rank, $P < 0.001$, $n \geq 15$).
C    Lysozyme IHC demonstrating increased lysozyme-positive cell numbers in tumours from Lgr5 Apc Huwe1 mice. Scale bars = 200 μm (low magnification), 100 μm (high-magnification inset).
D    OLFM4 IHC demonstrating expanded stem cell population in Lgr5 Apc Huwe1 tumours. Scale bars = 200 μm (low magnification), 100 μm (high-magnification inset).

express low levels of *HUWE1* expression respond better to adjuvant chemotherapy treatment than those with high levels of *HUWE1*. Due to the relatively high prevalence of *HUWE1* mutations in a number of cancer types, this may indicate an important therapeutic window for their treatment.

One unexpected finding was that cells in the small intestine tolerated accumulation of DNA damage following *Huwe1* deletion without undergoing apoptosis. This appears in conflict with previous reports of sensitivity of intestinal cells to numerous DNA-damaging agents (Phesse *et al*, 2014). Using genetic deletion of a single *Mcl1* allele, we found that protection against cell death is conferred by increased levels of the anti-apoptotic protein MCL1, a previously

described HUWE1 target (Zhong *et al*, 2005). Thus, tumour cell survival in the context of increased DNA damage following *Huwe1* deletion is partially dependent on increased levels of MCL1. Currently, there is much excitement of the use of BCL2 family inhibitors in hematopoietic malignancies such as CLL. Thus far, there has not been such strong preclinical evidence that inhibition of BCL2 family proteins will be effective for the treatment of late-stage epithelial cancers. Recent studies have shown that intestinal stem cells have high levels of BCL2 and that loss of BCL2 could slow tumourigenic from Lgr5 stem cells (van der Heijden *et al*, 2016). Our work here would suggest increased expression of MCL1 (a *bona fide* HUWE1 target) in *Huwe1*-deficient cells might protect these

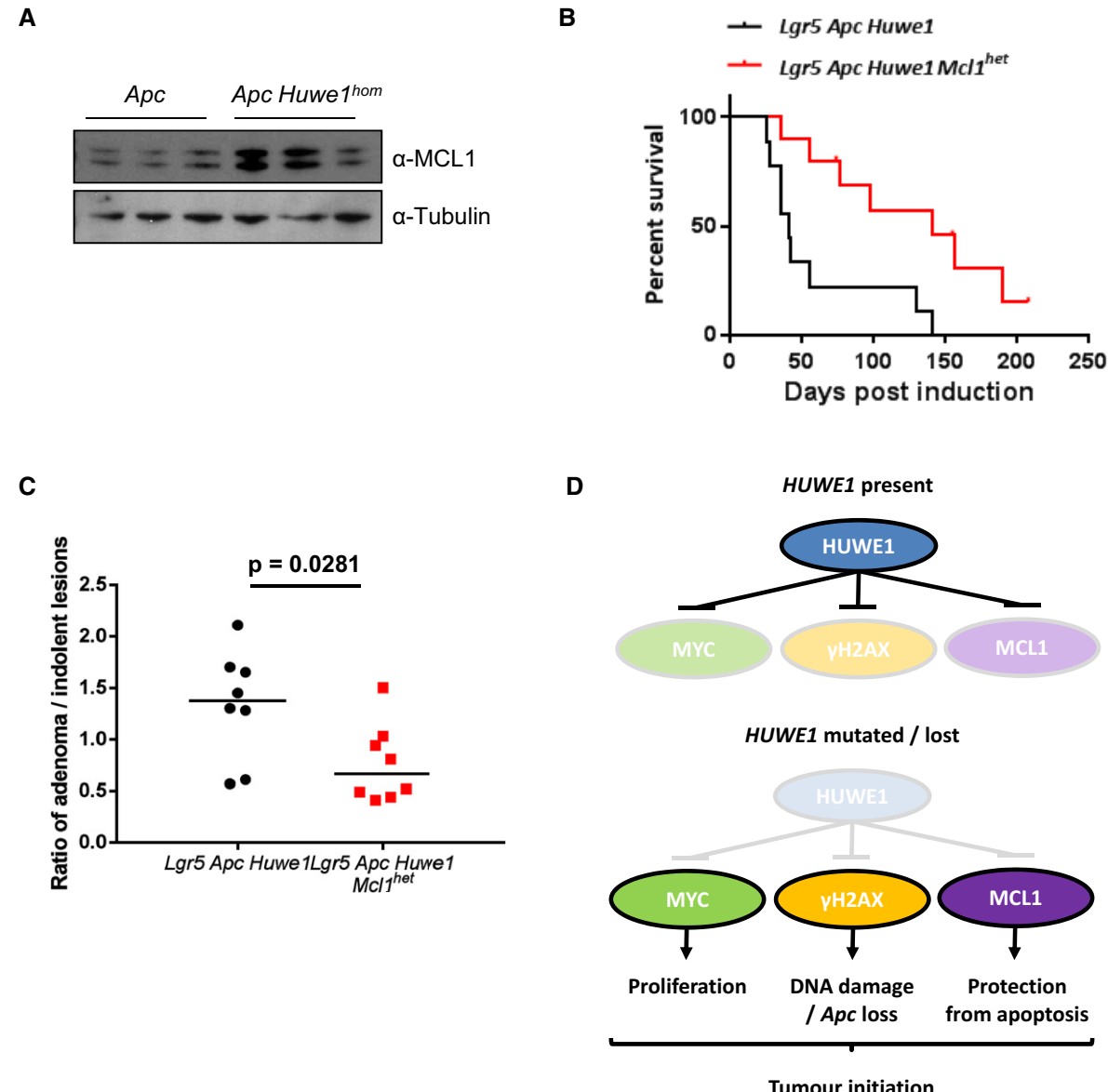

**Figure 8.  *Huwe1*-deficient tumours tolerate high levels of DNA damage due to elevated MCL1.**

A   MCL1 Western blot in protein extracts from *Vil Apc* and *Vil Apc Huwe1^hom* tumours. Levels of MCL1 protein are significantly increased in tumours lacking HUWE1 (Mann–Whitney, *P* = 0.04, *n* = 3).

B   Kaplan–Meier survival plot of cohorts of induced *Lgr5 Apc Huwe1* and *Lgr5 Apc Huwe1 Mcl1^het* mice. Deletion of one copy of *Mcl1* led to a significant increase in survival of these animals (log rank, *P* = 0.0052, *n* = 9 versus 10).

C   Ratio of adenoma/indolent lesions observed in *Lgr5 Apc Huwe1* and *Lgr5 Apc Huwe1 Mcl1^het* mice. Note the decreased ratio of adenoma/indolent lesions observed upon *Mcl1* deletion (Mann–Whitney, *P* = 0.0281, *n* = 8 versus 8).

D   Schematic outlining the tumour-suppressive role of HUWE1. When HUWE1 is expressed, the amounts of MYC, γ-H2AX and MCL1 are maintained at low levels. Following *HUWE1* mutation or loss MYC, γ-H2AX and MCL1 levels are increased leading to increased levels of proliferation and DNA damage. This leads to loss of *Apc* and tumour initiation. Increased MCL1 is critical to protect these cells from apoptosis induced by the high levels of DNA damage.

Source data are available online for this figure.

cells from death as haploinsufficiency for MCL1 markedly slowed tumourigenic. Currently, MCL1 inhibitors are in development and compounds are entering phase 1 trials. Thus, HUWE1 mutations might allow enrichment for responders to MCL1 inhibition either alone or in combination with other BCL2 family inhibitors or chemotherapy.

Overall, we find that loss of *Huwe1* function drives a potent tumour initiation phenotype in the context of *Apc* heterozygosity. This phenotype is mediated by a number of key pro-oncogenic events including increased MYC stability, elevated DNA damage and increased levels of MCL1 promoting stem cell transformation and protection from cell death (Fig 8D). Together, our work defines

a critical tumour suppressor pathway and provides the basis for further dissection of its role in CRC.

# Materials and Methods

## Mouse experiments

All animal experiments were performed under UK Home Office regulations (licence 70/8646) which underwent local ethical review at Glasgow University prior to being carried out. The backgrounds of mice used were mixed (50% C57BL/6J, 50% S129). Mice of both sexes were used during these studies at approximately equal ratios except for experiments analysing *Huwe1* heterozygous deletion where only females were used. Animals were bred in positively pressurised IVC caging, and all animal handling took place in change station or CAT11 hoods. Cages were autoclaved prior to use, with irradiated diet and 0.1-µm filter drinking water supplied by Hydropac. Animal holding rooms are supplied with HEPA-filtered air and the rooms are positively pressurised in relation to the other parts of the facility. For experimentation, animals were caged in conventional cages. These cages were autoclaved prior to use, with irradiated diet and autoclaved water supplied in bottles. Details of animal numbers used for each experiment are included in the relevant figure legend, and those were generally > 10 for tumour survival experiments (both control and experimental groups) and > 3 for short-term (up to 14 days) experiments. Mice were induced between 6 and 8 weeks of age and when weighing over 20 g. Genetic alleles used throughout this study were as follows: *vil-Cre-ER^{T2}* (el Marjou *et al*, 2004), *Apc^{fl}* (Shibata *et al*, 1997), *Huwe1^{fl}* (Zhao *et al*, 2008), *Myc^{fl}* (de Alboran *et al*, 2001), *Pten^{fl}* (Suzuki *et al*, 2001), *Mcl1^{fl}* (Opferman *et al*, 2003), *Lgr5-cre-ER^{T2}* (Barker *et al*, 2007) and *Ah-cre-ER^{T}* (Kemp *et al*, 2004). No randomisation was used, and for scoring experiments, blinding of sample genotype to the scorer was carried out. The experimental unit was designated as single animals. Mice were housed (DETAILS). Recombination in *vil-Cre-ER^{T2}* tumour models was induced using a single intraperitoneal (IP) injection of 80 mg/kg tamoxifen. Recombination in *vil-Cre-ER^{T2}* short-term models (days 4 and 14) was induced using a single IP injection of 80 mg/kg tamoxifen for two consecutive days. Recombination in *Lgr5-cre-ER^{T2}* tumour models was induced with a single IP injection of 120 mg/kg tamoxifen. Recombination in *Ah-cre-ER^{T}* tumour models was induced with a single IP injection of 80 mg/kg tamoxifen and 80 mg/kg β-naphthoflavone for four consecutive days. For cisplatin treatment, tumour-bearing mice were injected with 7.5 mg/kg cisplatin IP and sacrificed 6 h post-injection. For proliferation analysis, mice were injected with 250 µl of BrdU (Amersham Biosciences) 2 h before being sacrificed.

## Immunohistochemistry

We used standard immunohistochemistry techniques during this study. Details of primary antibodies and concentrations used can be found in Appendix Supplementary Methods. We performed staining on at least three mice of each genotype. Following blind scoring, representative images were selected for each scoring.

## Tissue sample scoring

Tissue samples were scored following various immunohistochemical staining. For normal tissue, the number of positive cells per crypt or half-crypt was scored where appropriate. At least three mice were used for each genotype. For tumour tissue, at least 20 images were captured at 40× magnification. The percentage of positive cells was scored for each image and the values averaged. At least three mice were used for each genotype.

## UbcH7 pulldown

*In vitro* binding assays were performed with the HUWE1-HECT domain (amino acids 4,015–4,374 of human Huwe1) produced as glutathione S-transferase (GST) fusion proteins. GST–HECT fusion proteins were purified for wild-type Huwe1, the catalytically inactive Huwe1 mutant C4341A (C/A) and the two human colorectal carcinoma-specific mutants (R4082H and K4204del). The E2 subunit UbcH7 was produced as a His6 fusion protein and eluted from Ni-NTA Agarose beads (Qiagen). The binding reactions were performed as described (Zhao *et al*, 2008), and the fraction of bound His-UbcH7 was measured by anti-histidine Western blot.

## Microarray analysis

One microgram of total intestinal RNA was reverse-transcribed to cDNA and hybridised to Affymetrix Mouse Genome 430 2.0 microarrays. CEL files of six samples were normalised and analysed in Partek Genomics Suite software. RMA normalisation and log2 transformation of the data were followed by differential expression analysis using ANOVA and *post hoc* linear contrasts between all pairs of experimental conditions. Multiple test corrections were carried out for *P*-values calculated. The fold change values for ranking genes of interest were then considered (Appendix Table S1). Gene set enrichment analysis was performed with chi-squared test with Yates's correction.

## Mass spectrometry analysis

Tissue samples were lysed in 2% SDS by sonication. Lysates were washed, reduced, alkylated, digested with trypsin and analysed on a Q-Exactive mass spectrometer as previously reported (Farrell *et al*, 2014). Proteins were identified and quantified by label-free quantification (LFQ) in the MaxQuant software suite (Cox *et al*, 2014) by searching against the mouse Uniprot database, with carbamylation of cysteines as a fixed modification and N-terminal acetylation and methionine oxidation as variable modifications. The LFQ values were transformed (log2) and grouped, 0 values imputed (normal distribution shifted 2π) and statistically distinct protein groups identified (permutation-based FDR, 0.05) by using the Perseus software suite (Tyanova *et al*, 2016).

## *HUWE1* expression analysis in human patients

We assembled an integrated database of colon cancer patient samples measured by Affymetrix HGU133A, HGU133Aplus2 and HGU133Av2 gene chips by employing the keywords "colon",

**The paper explained**

**Problem**

Cancer sequencing efforts have identified a large number of genes mutated at low frequency whose function during tumour initiation and growth is unknown. It is important to know whether these mutated genes play a role in how tumours develop and respond to treatments as this could guide how we choose to treat patients with these mutations.

**Results**

We identify a key role during colorectal tumour initiation for a gene called *HUWE1* that is mutated in 7–15% of colorectal cancer cases. Using animal cancer models, we demonstrate that *Huwe1* is a key tumour suppressor gene whose loss drives increased DNA damage. Importantly, this increased DNA damage phenotype sensitises *Huwe1*-deficient tumours to treatment with DNA-damaging agents and depletion of the anti-apoptotic protein MCL1.

**Impact**

This work demonstrates that infrequently mutated genes do play a functional role in colorectal cancer development. Importantly, by dissecting the mechanism via which *Huwe1* suppresses colonic tumourigenic, we identify a potential vulnerability of these tumours to DNA-damaging agents and anti-apoptotic inhibitors.

"cancer", "GPL96", "GPL571" and "GPL570" (respective platform accession numbers for each above gene chip) in NCBI GEO (http://www.ncbi.nlm.nih.gov/geo/). Only studies presenting raw data, clinical data including survival length, and at least 30 patients were included. Redundant samples ($n = 777$) were identified using the ranked expression of all genes and removed from the final combined database. The raw files were MAS5-normalised in the R environment using the Affy Bioconductor library (http://www.bioconductor.org). In the final analysis, samples with and without chemotherapy ($n = 244$ and $n = 318$, respectively) were analysed separately.

Further experimental details are provided in Appendix Supplementary Methods.

**Expanded View** for this article is available online.

## Acknowledgements

All authors are supported by Cancer Research UK. K.B.M. is funded by an AICR grant, a University of Edinburgh Chancellor's Fellowship and a Cancer Research UK Career Development Fellowship. The research leading to these results has received funding from the European Union Seventh Framework Programme FP7/2007-2013 under grant agreement number 278568. Thank you to histology, microscopy and the BSU for enabling this work to be performed.

## Author contributions

OJS, KBM, PC, MCH, AI, AL, DJA, FC, AVK and AM conceived and designed the project. KBM, PC, MR, JW, BGy, SP, EM, BG, EB and LV performed the experiments and analysed the data. KBM, OJS, PC, AI, AL, DJA and MR interpreted the data. KBM, OJS, PC, MCH, AI, AL and DJA wrote the manuscript.

## Conflict of interest

The authors declare that they have no conflict of interest.

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
