## [Review Process File · EMBO Molecular Medicine]

HUWE1 is a critical colonic tumour suppressor gene that prevents MYC signalling, DNA damage accumulation and tumour initiation

Kevin B. Myant, Patrizia Cammareri, Michael C. Hodder, Jimi Wills, Alex Von Kriegsheim, Balázs Gyórfy, Mamun Rashid, Simona Polo, Elena Maspero, Lynsey Vaughan, Basanta Gurung, Evan Barry, Angeliki Malliri, Fernando Camargo, David J. Adams, Antonio Iavarone, Anna Lasorella, and Owen J. Sansom

Corresponding author: Owen Sansom, Cancer Research UK Beatson Laboratories

Review timeline:

Submission date:	09 June 2016
Editorial Decision:	14 July 2016
Revision received:	20 October 2016
Editorial Decision:	04 November 2016
Revision received:	15 November 2016
Accepted:	21 November 2016

Transaction Report:

Editor: Roberto Buccione

1st Editorial Decision

14 July 2016

Thank you for the submission of your manuscript to EMBO Molecular Medicine. We are sorry that it has taken longer than we would have liked to get back to you on your manuscript.

As you will see the three reviewers are very positive, although they do raise some concerns on your manuscript, which I would basically summarise in 1) the need for further understanding/unravelling of the impact of HUWE1 on Wnt signalling and the other pathways and additional experimentation to crystallize some of the conclusions and 2) make a better case for human relevance.

The requests appear sound and aimed towards improvement of the quality and impact of the message. In brief, while publication of the paper cannot be considered at this stage, we would be pleased to consider a revised submission, with the understanding that the Reviewers' concerns must be addressed including further experimentation. Eventual acceptance of the manuscript will entail a second round of review.

We remind you that it is EMBO Molecular Medicine policy to allow a single round of revision only and that, therefore, acceptance or rejection of the manuscript will depend on the completeness of your responses included in the next, final version of the manuscript.

As you know, EMBO Molecular Medicine has a "scooping protection" policy, whereby similar

findings that are published by others during review or revision are not a criterion for rejection. However, I do ask you to get in touch with us after three months if you have not completed your revision, to update us on the status. Please also contact us as soon as possible if similar work is published elsewhere.

Please note that EMBO Molecular Medicine now requires a complete author checklist (<http://embomolmed.embopress.org/authorguide#editorial3>) to be submitted with all revised manuscripts. Provision of the author checklist is mandatory at revision stage; The checklist is designed to enhance and standardize reporting of key information in research papers and to support reanalysis and repetition of experiments by the community. The list covers key information for figure panels and captions and focuses on statistics, the reporting of reagents, animal models and human subject-derived data, as well as guidance to optimise data accessibility.

Please carefully adhere to our guidelines for authors (<http://embomolmed.embopress.org/authorguide>) to accelerate manuscript processing in case of acceptance.

I look forward to seeing a revised form of your manuscript as soon as possible.

***** Reviewer's comments *****

Referee #1 (Comments on Novelty/Model System):

This is a well thought out and technically sound study using mouse models to study the function of Huwe1 in intestinal tumorigenesis.

Referee #1 (Remarks):

The ubiquitin ligase Huwe1 is frequently mutated in colon cancer, but how loss of this gene promotes tumorigenesis is still poorly understood. In this study, the authors confirm earlier work showing that Huwe1 acts as a negative regulator of Wnt signaling. However, they also provide evidence that this role in Wnt signaling is not sufficient to explain the effect of Huwe1 on intestinal cancer formation. Instead, they show that the function of Huwe1 in regulating Myc, DNA damage control and the anti-apoptotic protein Mcl1 is important for tumor initiation.

Although these data are compelling, there are still some questions on the relationship between the effects of Huwe1 on Wnt signaling and these other pathways. For example, the authors show that loss of Huwe1 leads to increased expression of known Wnt target genes such as Ephb3, but they do not observe an effect on Myc expression, although this is one of the key Wnt target genes in the intestine. What is the explanation for this?

Figure S5 shows that nuclear beta-catenin levels are higher in Apc Huwe1 double mutant adenomas than in Apc single mutant adenomas. Does this suggest that loss of Huwe1 promotes beta-catenin stabilization downstream of Apc?

Finally, it would be informative to know whether the increased survival in the Apc Huwe1 Mcl1 mutant mice is correlated with a lower tumor number, as would be expected.

Referee #2 (Comments on Novelty/Model System):

This study examines the role of Huwe1 in animal models for colon cancer. The authors use several genetic models to address different questions about Huwe1 function. Although this can be useful for understanding the role of Huwe1 in different situations, it raises questions about how common does Huwe1 contribute to the disease.

Referee #2 (Remarks):

In this manuscript, the authors use different genetic models to address the role of Huwe1 in colon

cancer. Loss of Huwe1 is associated with increased levels of DNA damage, increased tumor cell growth and reduced apoptosis. The reason for the increased levels of DNA damage is not clear. The authors also report elevated levels of putative Huwe1 substrates such as Myc and Mcl1 although the changes in levels are quite small. This study will be significantly improved if the authors can determine whether those proteins are indeed substrates of Huwe1 in their models. In addition, it is not clear why only upon loss of Huwe1 do tumor cells become dependent on Myc for growth or Mcl1 for survival.

Referee #3 (Comments on Novelty/Model System):

Traditionally, many efforts in cancer research are focused at understanding the biology of driver genes (frequently mutated) in cancer development and progression. Currently, genome-wide sequencing efforts of patient-derived tumor samples provide increasing amounts of data, thereby continuously increasing the resolution of mutational frequencies of low-abundant mutant genes. As a result, these infrequent mutant genes can be clearly separated from the 'mutational noise'. However, the relevance of these low abundant mutant genes on cancer initiation and progression are mostly unknown.

Huwe1 is an example of such a low abundant mutated gene (up to 15% in colorectal cancer). In light of a recent publication, Huwe1 deficiency has been shown to elevate numbers of intestinal tumors. Importantly, this manuscript clearly discriminates between the Huwe1 mechanisms of tumor growth (MYC) and tumor initiation (DNA damage).

Via a tour de force using mouse genetics the authors convincingly show that loss of Huwe1 within an APC heterozygous background dramatically increases the number of colorectal tumors and thereby shortens the lifespan of the mice.

The underlying mechanism following Huwe1 depletion is three-fold:

- 1) Stimulation of tumor growth/proliferation (but not initiation) via MYC accumulation.
- 2) Accumulation of DNA damage (thereby accelerating LOH for APC and increasing tumor initiation).
- 3) Reduced sensitivity for apoptotic signals due to stabilization of anti-apoptotic protein MCL1.

This is a very important paper that describes the *in vivo* effects of Huwe1 deficiency during initiation and growth of colorectal tumors using powerful *in vivo* mouse models. The manuscript shows robust findings and a lengthy discussion regarding the previous conflicting and controversial findings regarding the tumorigenic role of Huwe1.

Comments:

- From the data provided in this manuscript, it is unclear whether increased levels of H2AX is a cause or consequence of accumulated DNA damage. It would be a pre if the authors can show a direct link between Huwe1 and H2AX levels in intestinal tissue, since increased levels of DNA damage is used to explain the elevated levels of tumor initiation that is at the heart of the phenotype.

- Regarding the fact that Huwe1 is mutated in multiple types of cancer, did the authors made any observations about tumor initiation and growth in the stomach (Lgr5-driven) or colon (Lgr5 or Villin-driven) after Huwe1 and APC depletion?

- The fact that Huwe1 deficient tumors displayed increased sensitivity towards cisplatin treatments is of particular importance with respect to human healthcare. Can the authors strengthen these lines of evidence using clinical patient data with respect to drug response towards cisplatin (or any other chemotherapy) and Huwe1 status?

- Regarding the controversy of Paneth cell mislocalization versus lineage commitment/ectopic lysozyme expression. Can the authors perform additional qPCRs or *in situ* hybridizations for Paneth cell transcripts such as MMP7 or Defensins to strengthen their arguments that mature Paneth cells are not mislocalized?

Minor comments:

- Typo at the top of page 7. 'These data indicate that HUWE1 controls MYC protein abundance in both normal ...'

- Figure legend 3A. I assumed this western blot is of Huwe1 levels in 'normal' homeostasis and not in tumors?

Referee #3 (Remarks):

This is a very important paper that describes the in vivo effects of Huwe1 deficiency during initiation and growth of colorectal tumors using genetic mouse models. The data is of high quality. Most importantly considering a recent publication about Huwe1 deficiency in the intestine, this manuscript clearly discriminates between Huwe1 mechanisms regarding tumor growth (MYC) and initiation (DNA damage).

1st Revision - authors' response

20 October 2016

Referee #1 (Comments on Novelty/Model System):

This is a well thought out and technically sound study using mouse models to study the function of Huwe1 in intestinal tumorigenesis.

Referee #1 (Remarks):

The ubiquitin ligase Huwe1 is frequently mutated in colon cancer, but how loss of this gene promotes tumorigenesis is still poorly understood. In this study, the authors confirm earlier work showing that Huwe1 acts as a negative regulator of Wnt signaling. However, they also provide evidence that this role in Wnt signaling is not sufficient to explain the effect of Huwe1 on intestinal cancer formation. Instead, they show that the function of Huwe1 in regulating Myc, DNA damage control and the anti-apoptotic protein Mcl1 is important for tumor initiation.

Although these data are compelling, there are still some questions on the relationship between the effects of Huwe1 on Wnt signaling and these other pathways. For example, the authors show that loss of Huwe1 leads to increased expression of known Wnt target genes such as Ephb3, but they do not observe an effect on Myc expression, although this is one of the key Wnt target genes in the intestine. What is the explanation for this?

The reviewer makes an important point and we believe the reason for this is that *Huwe1* deletion only has a very mild impact on Wnt signalling. For example, we only see a subset of Wnt target genes activated and see no evidence for increased nuclear beta-catenin levels in either normal or tumorigenic tissue. With this in mind we have tempered our conclusions around the role of HUWE1 in Wnt signalling to emphasise that its role in Wnt signalling regulation is minor. Text has been added on pages 6, 10 and 11 to further clarify this. Importantly, this matter does not impact on the main conclusion of this study which is that the effects of *Huwe1* deletion on Wnt signalling are clearly insufficient to explain the striking tumour initiation phenotype we observe.

Figure S5 shows that nuclear beta-catenin levels are higher in *Apc* *Huwe1* double mutant adenomas than in *Apc* single mutant adenomas. Does this suggest that loss of Huwe1 promotes beta-catenin stabilization downstream of *Apc*?

This is not something we believe to be the case but close analysis of the figure does indicate a potential change in nuclear beta-catenin levels in *Huwe1* deficient tumours. To address this more carefully we have investigated as follows. First, we quantified both number and staining intensity of nuclear beta-catenin across multiple tumours from mice deleted for *Apc* or both *Apc* and *Huwe1*. This analysis conclusively shows there is no difference in nuclear beta-catenin levels between these two genotypes (Figure S2). Given this quantification showed no difference we have now replaced the figure. Second, we carried out expression analysis of Wnt target genes on *Apc* deficient tissue with additional *Huwe1* deletion. These analyses find no evidence for increased Wnt target gene expression in *Apc* deficient tissue following *Huwe1* loss (Figure

S3F). Again, this has been clarified in the text on pages 5 and 11. Like the above point, again this does not impact on our main conclusions that the further deregulation of Wnt signalling by HUWE1 is not the reason for faster tumourigenesis.

Finally, it would be informative to know whether the increased survival in the *Apc Huwe1 Mcl1* mutant mice is correlated with a lower tumor number, as would be expected.

We have included tumour scoring data from this experiment and find that adenoma number is slightly lower (not significant) in *Mcl1* heterozygous animals. In addition to this, the number of indolent lesions is slightly higher and the ratio of these is significantly changed. These data are included in Fig 8 and Fig S6.

Referee #2 (Comments on Novelty/Model System):

This study examines the role of Huwe1 in animal models for colon cancer. The authors use several genetic models to address different questions about Huwe1 function. Although this can be useful for understanding the role of Huwe1 in different situations, it raises questions about how common does Huwe1 contribute to the disease.

Referee #2 (Remarks):

In this manuscript, the authors use different genetic models to address the role of Huwe1 in colon cancer. Loss of Huwe1 is associated with increased levels of DNA damage, increased tumor cell growth and reduced apoptosis. The reason for the increased levels of DNA damage is not clear. The authors also report elevated levels of putative Huwe1 substrates such as Myc and Mcl1 although the changes in levels are quite small. This study will be significantly improved if the authors can determine whether those proteins are indeed substrates of Huwe1 in their models. In addition, it is not clear why only upon loss of Huwe1 do tumor cells become dependent on Myc for growth or Mcl1 for survival.

We agree that showing an interaction between HUWE1 and MYC and MCL1 in intestinal epithelium would be beneficial. We have attempted to do this by carrying out CoIP experiments in protein extracts from mouse small intestine. Unfortunately, we have been unable to successfully CoIP these proteins from this tissue. We believe this is due to technical issues regarding the antibodies used working in mouse intestinal samples as we have not been able to conclusively demonstrate single IPs of any of them. *In vivo* CoIP experiments are technically challenging and although beneficial we do not believe they are necessary for our paper as the targets we are investigating are HUWE1 targets validated by multiple groups, with HUWE1 and MYC interactions reported by several groups (Adhikary et al., 2005; Inoue et al., 2013) and the same for HUWE1 and MCL1 interactions (Pervin et al., 2011; Zhong et al., 2005). Therefore as we show protein upregulation and a genetic interaction of the pathways we think that this provides robust evidence that these are important targets downstream of HUWE1 for the phenotypes we see.

Regarding why tumour cells become dependent on MYC for growth and MCL1 for survival following *Huwe1* deletion we believe this is due to the following. Firstly, tumour cells do not become dependent on MYC for growth following *Huwe1* deletion (the *Apc Pten Huwe1 Myc* tumours proliferate as well as *Apc Pten* tumours – see Fig S4b). It is the *increased* proliferation we observe upon *Huwe1* deletion that is dependent on MYC. Secondly, we believe *Huwe1* deficient tumours are dependent on MCL1 for survival due to the increased DNA damage burden we observe following *Huwe1* loss. Thus, these cells become highly dependent on anti-apoptotic pathways due to the increased DNA damage and associated stress. We have clarified both points in the text (page 10 for MYC, page 11/12 for MCL1).

Referee #3 (Comments on Novelty/Model System):

Traditionally, many efforts in cancer research are focused at understanding the biology of driver genes (frequently mutated) in cancer development and progression. Currently, genome-wide sequencing efforts of patient-derived tumor samples provide increasing amounts of data, thereby continuously increasing the resolution of mutational frequencies of low-abundant mutant genes. As a result, these infrequent mutant genes can be clearly separated from the 'mutational noise'. However, the relevance of these low abundant mutant genes on cancer initiation and progression are

mostly unknown.

Huwe1 is an example of such a low abundant mutated gene (up to 15% in colorectal cancer). In light of a recent publication, Huwe1 deficiency has been shown to elevate numbers of intestinal tumors. Importantly, this manuscript clearly discriminates between the Huwe1 mechanisms of tumor growth (MYC) and tumor initiation (DNA damage).

Via a tour de force using mouse genetics the authors convincingly show that loss of Huwe1 within an APC heterozygous background dramatically increases the number of colorectal tumors and thereby shortens the lifespan of the mice.

The underlying mechanism following Huwe1 depletion is three-fold:

- 1) Stimulation of tumor growth/proliferation (but not initiation) via MYC accumulation.
- 2) Accumulation of DNA damage (thereby accelerating LOH for APC and increasing tumor initiation).
- 3) Reduced sensitivity for apoptotic signals due to stabilization of anti-apoptotic protein MCL1.

This is a very important paper that describes the *in vivo* effects of Huwe1 deficiency during initiation and growth of colorectal tumors using powerful *in vivo* mouse models. The manuscript shows robust findings and a lengthy discussion regarding the previous conflicting and controversial findings regarding the tumorigenic role of Huwe1.

Comments:

- From the data provided in this manuscript, it is unclear whether increased levels of H2AX is a cause or consequence of accumulated DNA damage. It would be a pre if the authors can show a direct link between Huwe1 and H2AX levels in intestinal tissue, since increased levels of DNA damage is used to explain the elevated levels of tumor initiation that is at the heart of the phenotype. **We have analysed H2AX levels by Western blot and find they are elevated following *Huwe1* deletion. In addition we have carried out unbiased mass spectrometry analysis of the same tissue which also identified increased levels of H2AX. Thus, we have shown using two independent techniques that loss of *Huwe1* leads to increased levels of H2AX protein which confirms data already published (Atsumi et al., 2015). We have not been able to show a direct interaction between Huwe1 and H2AX proteins in intestinal tissue due to technical difficulties in carrying out *in vivo* CoIPs.**

- Regarding the fact that Huwe1 is mutated in multiple types of cancer, did the authors made any observations about tumor initiation and growth in the stomach (*Lgr5*-driven) or colon (*Lgr5* or Villin-driven) after Huwe1 and APC depletion?

We also observed increased tumour initiation in the colons of mice from the *VilCre* driven tumour model. We have included this data in Fig S1. We did not observe any obvious stomach lesions in the *Lgr5* driven model, perhaps due to the dominance of the intestinal phenotype in this model. We agree that this would certainly be interesting to follow up more carefully in a future study.

- The fact that Huwe1 deficient tumors displayed increased sensitivity towards cisplatin treatments is of particular importance with respect to human healthcare. Can the authors strengthen these lines of evidence using clinical patient data with respect to drug response towards cisplatin (or any other chemotherapy) and Huwe1 status?

This is an excellent suggestion and we have attempted to link chemotherapy response to *HUWE1* mutation status in human patients. Unfortunately, the data from TCGA colorectal project does not include data on clinical response to chemotherapy so we cannot do this. On the other hand we have analysed data on *HUWE1* expression levels and how patients respond to treatment. This data is included in Figure 6 and shows that patients with low expression of *HUWE1* have significantly better survival if they have undergone chemotherapy. This is not the case in patients not treated with chemotherapy. Together, this correlates with our hypothesis that low levels of *HUWE1* lead to accumulation of DNA damage and subsequent sensitivity to chemotherapeutic agents.

- Regarding the controversy of Paneth cell mislocalization versus lineage commitment/ectopic lysozyme expression. Can the authors perform additional qPCRs or *in situ* hybridizations for Paneth cell transcripts such as MMP7 or Defensins to strengthen their arguments that mature Paneth cells

are not mislocalized?

We have included IHC for MMP7 which clearly shows no mislocalisation from the crypt base strongly arguing against functional Paneth cell mislocalisation (Fig 2).

Minor comments:

- Typo at the top of page 7. 'These data indicate that HUWE1 controls MYC protein abundance in both normal ...'

This has been corrected.

- Figure legend 3A. I assumed this western blot is of Huwe1 levels in 'normal' homeostasis and not in tumors?

Yes, this is showing MYC levels in normal homeostasis and has now been corrected.

Referee #3 (Remarks):

This is a very important paper that describes the in vivo effects of Huwe1 deficiency during initiation and growth of colorectal tumors using genetic mouse models. The data is of high quality. Most importantly considering a recent publication about Huwe1 deficiency in the intestine, this manuscript clearly discriminates between Huwe1 mechanisms regarding tumor growth (MYC) and initiation (DNA damage).

References

- Adhikary, S., Marinoni, F., Hock, A., Hulleman, E., Popov, N., Beier, R., Bernard, S., Quarto, M., Capra, M., Goettig, S., *et al.* (2005). The ubiquitin ligase HectH9 regulates transcriptional activation by Myc and is essential for tumor cell proliferation. *Cell* *123*, 409-421.
- Atsumi, Y., Minakawa, Y., Ono, M., Dobashi, S., Shinohe, K., Shinohara, A., Takeda, S., Takagi, M., Takamatsu, N., Nakagama, H., *et al.* (2015). ATM and SIRT6/SNF2H Mediate Transient H2AX Stabilization When DSBs Form by Blocking HUWE1 to Allow Efficient gammaH2AX Foci Formation. *Cell Rep* *13*, 2728-2740.
- Inoue, S., Hao, Z., Elia, A.J., Cescon, D., Zhou, L., Silvester, J., Snow, B., Harris, I.S., Sasaki, M., Li, W.Y., *et al.* (2013). Mule/Huwe1/Arf-BP1 suppresses Ras-driven tumorigenesis by preventing c-Myc/Miz1-mediated down-regulation of p21 and p15. *Genes Dev* *27*, 1101-1114.
- Pervin, S., Tran, A., Tran, L., Urman, R., Braga, M., Chaudhuri, G., and Singh, R. (2011). Reduced association of anti-apoptotic protein Mcl-1 with E3 ligase Mule increases the stability of Mcl-1 in breast cancer cells. *British journal of cancer* *105*, 428-437.
- Zhong, Q., Gao, W., Du, F., and Wang, X. (2005). Mule/ARF-BP1, a BH3-only E3 ubiquitin ligase, catalyzes the polyubiquitination of Mcl-1 and regulates apoptosis. *Cell* *121*, 1085-1095.

2nd Editorial Decision

04 November 2016

Thank you for the submission of your revised manuscript to EMBO Molecular Medicine. We have now received the enclosed report from the Reviewer 3 who was asked to re-assess it. As you will see the reviewer is now globally supportive. As for the requests from the other reviewers, we have assessed your actions editorially. I am pleased to inform you that we will be able to accept your manuscript pending the following final amendments:

- 1) The manuscript must include a statement in the Materials and Methods identifying the institutional and/or licensing committee approving the experiments, including any relevant details (like how many animals were used, of which gender, at what age, which strains, if genetically modified, on which background, housing details, etc). We encourage authors to follow the ARRIVE guidelines for reporting studies involving animals. Please see the EQUATOR website for details: <http://www.equator-network.org/reporting-guidelines/improving-bioscience-research-reporting-the-arrive-guidelines-for-reporting-animal-research/>. Please make sure that ALL the above details are reported in the manuscript.

- 2) Please provide figures as individual files
- 3) EMBO Press journals encourage the inclusion of extra figures (supplementary) in the HTML version of the main manuscript (Expanded View). If you wish to take advantage of this please refer to our guidelines (<http://embomolmed.embopress.org/authorguide#expandedview>). If you wish instead to keep the Appendix format only, please add a ToC and include the Appendix tables in the same file. If you have any questions regarding this do not hesitate to contact us.
- 4) Please adapt the shape of the outlined area on Fig. 5d main images to the magnification inset.
- 5) Please move the following Material and Methods sub-sections to the main text: UbcH7 pulldown, Microarray Analysis, Mass spectrometry analysis, and HUWE1 expression analysis in human patients
- 6) We encourage the publication of source data, particularly for electrophoretic gels and blots, with the aim of making primary data more accessible and transparent to the reader. Would you be willing to provide a PDF file per figure that contains the original, uncropped and unprocessed scans of all or at least the key gels used in the manuscript? The PDF files should be labeled with the appropriate figure/panel number, and should have molecular weight markers; further annotation may be useful but is not essential. The PDF files will be published online with the article as supplementary "Source Data" files.
- 7) Every published paper includes a 'Synopsis' to further enhance discoverability. Synopses are displayed on the journal webpage and are freely accessible to all readers. They include a short standfirst as well as 2-5 one sentence bullet points that summarise the paper. Please provide the synopsis including the short list of bullet points that summarise the key NEW findings. The bullet points should be designed to be complementary to the abstract - i.e. not repeat the same text. We encourage inclusion of key acronyms and quantitative information. Please use the passive voice. Please attach this information in a separate file or send them by email, we will incorporate it accordingly. You are also welcome to suggest a striking image or visual abstract to illustrate your article. If you do please provide a jpeg file 550 px-wide x 400-px high.
- 8) Data described in submitted manuscripts should be deposited in a MIAME-compliant format with one of the public databases. We would therefore ask you to submit your microarray data to the ArrayExpress database maintained by the European Bioinformatics Institute for example. ArrayExpress allows authors to submit their data to a confidential section of the database, where they can be put on hold until the time of publication of the corresponding manuscript. Please see <http://www.ebi.ac.uk/arrayexpress/Submissions/> or contact the support team at arrayexpress@ebi.ac.uk for further information.

Please submit your revised manuscript within two weeks. I look forward to seeing a revised form of your manuscript as soon as possible.

***** Reviewer's comments *****

Referee #3 (Comments on Novelty/Model System):

This is a sound study using mouse models to study the function of Huwe1 in intestinal tumorigenesis. Minor comments have been addressed appropriately. Moreover, the current version has now a stronger link with clinical data.

2nd Revision - authors' response

15 November 2016

Authors made requested changes.

Corresponding Author Name: Prof Owen Sansom, Dr Kevin Myant

Manuscript Number: EMM-2016-06684-V2